# Chondroitin sulfate proteoglycans prevent immune cell phenotypic conversion and inflammation resolution via TLR4 in rodent models of spinal cord injury

Isaac Francos-Quijorna[1,6], Marina Sánchez-Petidier[2,8], Emily R. Burnside[1,7,8], Smaranda R. Badea[1,8], Abel Torres-Espin[3], Lucy Marshall[1], Fred de Winter [4], Joost Verhaagen[4,5], Victoria Moreno-Manzano [2] & Elizabeth J. Bradbury [1✉]

Chondroitin sulfate proteoglycans (CSPGs) act as potent inhibitors of axonal growth and neuroplasticity after spinal cord injury (SCI). Here we reveal that CSPGs also play a critical role in preventing inflammation resolution by blocking the conversion of pro-inflammatory immune cells to a pro-repair phenotype in rodent models of SCI. We demonstrate that enzymatic digestion of CSPG glycosaminoglycans enhances immune cell clearance and reduces pro-inflammatory protein and gene expression profiles at key resolution time points. Analysis of phenotypically distinct immune cell clusters revealed CSPG-mediated modulation of macrophage and microglial subtypes which, together with T lymphocyte infiltration and composition changes, suggests a role for CSPGs in modulating both innate and adaptive immune responses after SCI. Mechanistically, CSPG activation of a pro-inflammatory phenotype in pro-repair immune cells was found to be TLR4-dependent, identifying TLR4 signalling as a key driver of CSPG-mediated immune modulation. These findings establish CSPGs as critical mediators of inflammation resolution failure after SCI in rodents, which leads to prolonged inflammatory pathology and irreversible tissue destruction.

[1] King's College London, Regeneration Group, The Wolfson Centre for Age-Related Diseases, Institute of Psychiatry, Psychology & Neuroscience, Guy's Campus, London Bridge, London SE1 1UL, UK. [2] Neuronal and Tissue Regeneration Laboratory, Prince Felipe Research Center, Carrer d´Eduardo Primo Yúfera 3, 46012 Valencia, Spain. [3] Brain and Spinal Injury Center, Neurological Surgery, University of California San Francisco, San Francisco, USA. [4] Laboratory for Neuroregeneration, Netherlands Institute for Neuroscience, Royal Netherlands Academy of Sciences, Meibergdreef 47, 1105 BA Amsterdam, The Netherlands. [5] Center for Neurogenomics and Cognition Research, Neuroscience Campus Amsterdam, Vrije Universiteit Amsterdam, 1081HV Amsterdam, The Netherlands. [6] Present address: Immunometabolism and Inflammation Laboratory, Centro de Biología Molecular Severo Ochoa (CBMSO), Consejo Superior de Investigaciones Científicas (CSIC)-Universidad Autónoma de Madrid (UAM), 28049 Madrid, Spain. [7] Present address: Laboratory of Axonal Growth and Regeneration, German Center for Neurodegenerative Diseases (DZNE), Venusberg Campus 1/99, 53127 Bonn, Germany. [8] These authors contributed equally: Marina Sánchez-Petidier, Emily R. Burnside, Smaranda R. Badea. ✉email: elizabeth.bradbury@kcl.ac.uk

Spinal cord injury (SCI) typically leads to severe and permanent motor, sensory, and autonomic dysfunction due to the inability of the adult mammalian CNS to regenerate lost neurons and re-establish functional connections[1]. Several factors have been identified that influence this regeneration failure and one of the major contributors is the presence of inhibitory chondroitin sulfate proteoglycans (CSPGs), which accumulate in and around spinal injury scar tissue[2,3]. Much attention has been focused on therapeutic strategies to reduce the growth inhibitory properties of CSPGs, such as Chondroitinase ABC (ChABC), a bacterial enzyme which, through catabolism of CSPG glycosaminoglycan (GAG) side chains has been shown to increase spinal and brain plasticity and facilitate functional recovery following injury[4–9]. Thus, the role of CSPGs in restricting growth and neuroplasticity has been well-studied. However, new roles for CSPGs in modulating CNS pathology beyond neuronal growth inhibition are emerging. Recent evidence suggests that CSPGs play a key role in modulating immune cell responses in chronic inflammatory and demyelinating disorders of the CNS[10–14]. Given their abundant expression, proximity, and interactions with multiple reactive cell types after SCI[2], CSPGs may also be important contributors to the pathological neuroinflammatory response after SCI.

SCI triggers an aggressive immune response which is characterised by activation of resident microglia and recruitment of peripheral leukocytes to the site of injury[15–17]. While the inflammatory response plays a critical role in tissue protection and wound healing after injury[18–20], an effective resolution is a prerequisite for a return to homoeostasis. Inadequate resolution can lead to chronic pathological inflammation that causes greater damage and impaired tissue healing[19,21]. Inflammation resolution failure is particularly problematic after SCI[16,22,23] and the mechanisms which propagate inflammatory pathology and prevent immune cell clearance after SCI are poorly understood. We and others have recently identified a pro-inflammatory role for CSPGs after SCI, where they amplify the immune response and contribute to poor functional outcome[24–27]. However, the mechanisms regulating CSPG–immune interactions remain poorly understood. How they affect dynamic responses of innate and adaptive immune cells, and which bioactive mediators and signalling pathways are involved is still unknown. Here we demonstrate that CSPGs have prominent immunomodulatory effects which impede multiple aspects of the resolution of inflammation after SCI. Digestion of CSPGs by lentiviral ChABC attenuated the pro-inflammatory environment, led to enhanced clearance/reduced recruitment of microglia and peripheral myeloid cells from the lesion site and activated resolution of the inflammatory response, modulating the phenotype of microglial cells and monocyte/macrophages and T lymphocyte infiltration and composition, with effects predominant at day 7 after SCI (a critical time point for inflammation resolution). Spatially, CSPGs prevent immune cells at the injury core from converting to a pro-repair phenotype, which can be reversed by their digestion. Furthermore, we reveal TLR4 signalling as a key mechanism underlying CSPG-mediated modulation of the inflammatory signature after SCI. Using TLR4 inhibitors and TLR4 knockout mice we demonstrate that activation of TLR4 signalling by CSPGs prevents macrophages and microglia from converting to a pro-resolution phenotype. We propose this as a key mediator of failed inflammation resolution after SCI. These data reveal a new role for CSPGs in preventing inflammation resolution and suggests they are a critical mediator of non-resolving inflammatory pathology after SCI.

## Results

**CSPG digestion promotes enhanced clearance/reduced recruitment of immune cells after spinal cord injury.** To determine whether the failed resolution of inflammation after SCI is regulated by CSPGs, we used a lentiviral vector expressing chondroitinase ABC (LV-ChABC) to digest CSPGs in the injured rat spinal cord[24,28] and used flow cytometry to evaluate the dynamics of the major inflammatory cell types over 28 days post injury (dpi; study design, Fig. 1a, c). To confirm efficient ChABC expression in the injured spinal cord during the neuroinflammatory period, qPCR was used to measure ChABC mRNA in spinal cord tissue, confirming high ChABC transgene expression in LV-ChABC-treated animals at all post-injury time points examined (from 6 h to 14 days, Extended Data Fig. 1a, b). Following confirmation of ChABC gene expression in the spinal cord and that contusion injuries were of equal severity across treatment groups (Fig. 1b), we first assessed the effects of CSPG digestion on immune cell dynamics using manual gating flow cytometry (Extended Data Fig. 1c, d and c–e). CSPG digestion did not affect recruitment of neutrophils (CD45+, CD11b+, CD43+, RP1+; Extended Data Fig. 1c) into the contused spinal cord as neutrophil numbers were unaltered by LV-ChABC treatment at 1 dpi, the peak time point for neutrophil accumulation (Extended Data Fig. 1d) or 3 dpi (3 dpi vs. 7 dpi—Extended Data Fig. 5e). However, LV-ChABC treatment accelerated the clearance of neutrophils from the contused spinal cord, evident by the reduced resolution index parameter compared with the LV-GFP control group (Ri = 52.8 and 40.3 h for LV-GFP and LV-ChABC treatment, respectively; Extended Data Fig. 1d), leading to a significant reduction in neutrophil accumulation at 7 dpi in animals treated with LV-ChABC (Extended Data Fig. 1d and Extended Data Fig. 5e). We next studied whether CSPGs interfere with the recruitment of macrophages (CD45+high, CD11b+; Extended Data Fig. 1c) after SCI. Peripheral macrophage infiltration into the contused spinal cord was not significantly different at 1 dpi or 3 dpi, the peak of macrophage infiltration, after CSPG digestion (Fig. 1d and 3 dpi vs. 7 dpi—Extended Data Fig. 5c). However, macrophage clearance during the later resolution phase was enhanced by LV-ChABC treatment, with macrophage numbers significantly reduced at 7 dpi compared to LV-GFP treatment (Fig. 1d). LV-ChABC treatment also attenuated the number of microglial cells (CD45+medium/low, CD11b+; Extended Data Fig. 1c) during the resolution phase of the inflammatory response, compared with LV-GFP control treatment (Fig. 1e and 3 dpi vs. 7 dpi—Extended Data Fig. 5a). These data indicate that CSPGs play a key role in hindering the clearance of immune cells after SCI.

Having used manual-gating flow cytometry to study dynamics, we next sought to evaluate the effects of CSPG digestion on immune cell recruitment at a more granular level, using unbiased t-distributed stochastic neighbour embedding (t-SNE) analysis, at a key resolution time point (7 dpi). We designed a 16-colour antibody panel for fluorescence cytometry (Supplementary Table 1—Panel 2) that identified all the major leucocytes in the CNS, including lineage markers (CD45, CD11b, CD43, HIS48, CD45RA, CD3, CD4, CD8) and markers linked to activation and phenotype differentiation (CD68, CD86, MHC-II, iNOS, CD163, CD206, and Arg I). Single, live, CD45+ cells from spinal cords at 7 dpi were analysed and samples were two-dimensionally mapped using t-SNE. Based on t-SNE data, we identified 9 phenotypically distinct clusters (Fig. 1f) and generated a heat map showing the distinct lineage marker expression profiles for each cluster (Fig. 1g). Lineage marker expression characterised cluster-1 as microglial cells. We observed clear visual changes in the size of cluster 1 between treatment groups (Fig. 1h) and cell number analysis showed a significant reduction of microglial cells after ChABC treatment (Fig. 1j), providing strong corroboration of the results obtained by manual-gating flow cytometry (Fig. 1e). Thus, CSPG digestion significantly reduces microglial cell accumulation. We next characterised the impact of CSPG digestion on the recruited monocyte/macrophage population at 7 dpi, identified as cluster 2 and 3 by the high expression of CD45 and CD11b

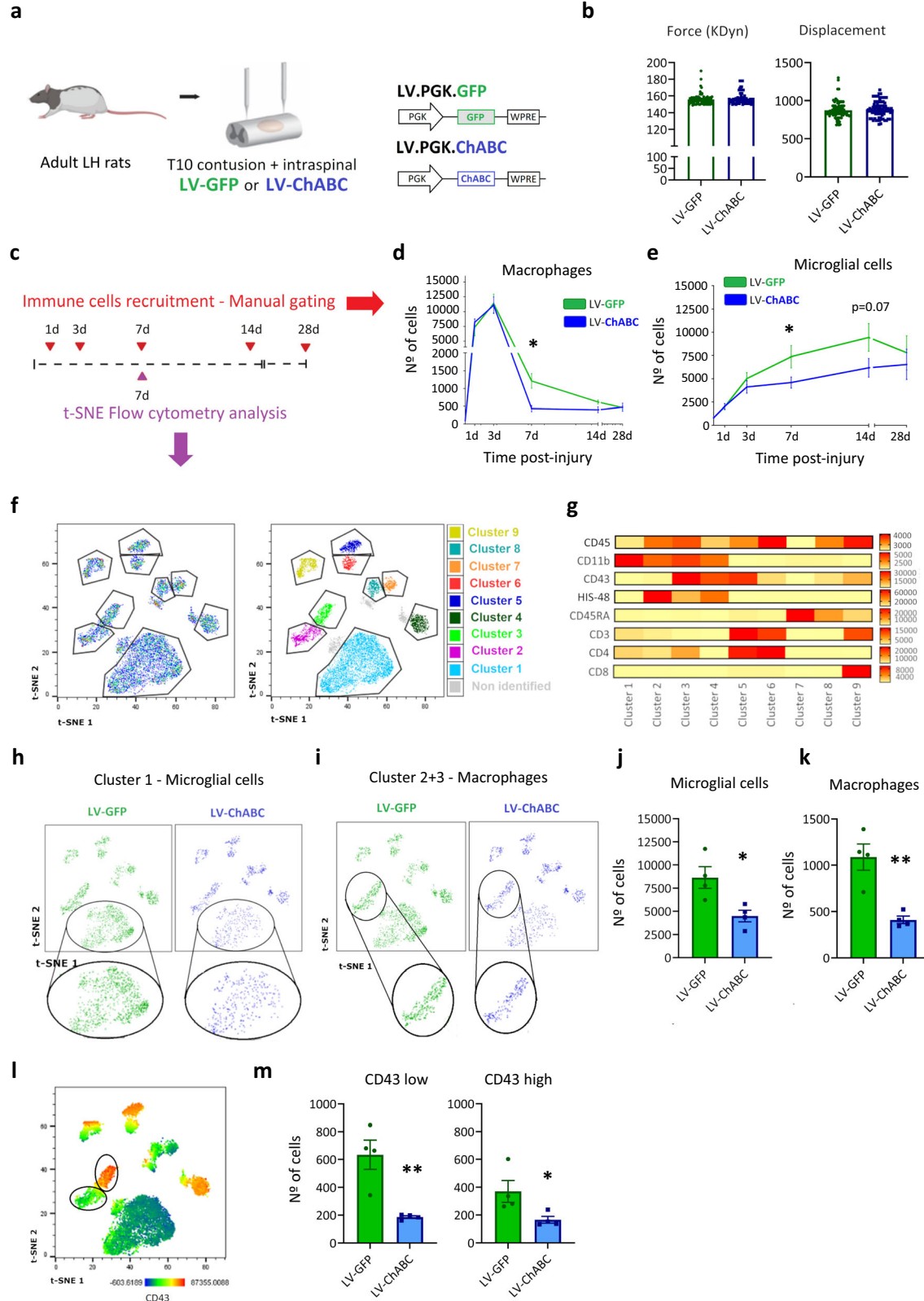

(Fig. 1f, g). In agreement with our observations with manual-gating flow cytometry analysis (Fig. 1d), CSPG digestion significantly reduced the monocyte/macrophage population within the injury epicentre at 7 dpi (Fig. 1i, k). Furthermore, t-SNE analysis identified two different monocyte/macrophage subsets based on differential expression of CD43, cluster 2 (CD43$^{low}$) and cluster 3 (CD43$^{high}$) (Fig. 1f, g, l). The effect of

CSPG digestion was analysed on these two subsets separately. As observed in the total monocyte/macrophage population, CSPG digestion produced a significant reduction in both monocyte/macrophage subsets, being more prominent in the CD43$^{low}$ population (Fig. 1m).

These data provide evidence that CPSG digestion enhances the clearance of peripheral myeloid cells and reduces microgliosis in

**Fig. 1 CSPG digestion enhances immune cell clearance after spinal cord injury. a, c** Experimental design of immune cell recruitment study. **b** Contusion device impact force and displacement measurements confirm reproducible and consistent injuries between treatment groups. Data were determined as normally distributed by the Shapiro–Wilk test and subsequently analysed using a two-tailed unpaired $t$ test. Data are shown as mean ± SEM. **d, e** Graphs showing quantification of innate immune cell recruitment, using manual-gated flow cytometry, following spinal cord injury at 1, 3, 7, 14 and 28 dpi with (LV-ChABC) or without (LV-GFP) CSPG digestion. *$p < 0.05$ versus control (LV-GFP) group. Results were assessed for normality using the Shapiro–Wilk test and analysed using a two-way ANOVA with Bonferroni's post hoc test ($n = 8$ per group at 1 dpi; $n = 11$ at 3 dpi; $n = 11$ at 7 dpi; $n = 6$ at 14 dpi; $n = 3$ at 28 dpi). Data are pooled from at least two independent experiments. Data are shown as mean ± SEM. **f** t-SNE flow cytometry analysis at 7dpi reveals the presence of 9 major CD45+ cell populations within the injured spinal cord. **g** Heat map showing the relative expression of extracellular markers in the 9 identified clusters. **h–m** CSPG digestion reduces the number of microglial cells and monocyte/macrophages at 7 dpi. **h** tSNE plot highlighting cluster 1 identified as microglia in LV-GFP (left) and LV-ChABC (right) treated rats. **i** tSNE plot highlighting clusters 2 and 3 identified as monocytes/macrophages in LV-GFP (left) and LV-ChABC (right) treated rats. **j** Graph showing the number of microglial cells within the injured spinal cord at 7 dpi. Microglial cell number is significantly reduced after CSPG digestion. **k** Graph showing the number of monocyte/macrophages within the injured spinal cord at 7 dpi. Monocyte/macrophage cell number is significantly reduced after CSPG digestion. **l, m** LV-ChABC treatment exhibits greater effects in CD43^low monocyte/ macrophages. **l** tSNE plots showing the expression of CD43 in the different clusters. Note that monocytes/macrophages are differentiated in two different subsets by CD43 relative expression. **m** Graph showing the number of CD43^high and CD43^low monocyte/macrophages within the injured spinal cord at 7 dpi. Both populations are significantly reduced after CSPG digestion. **j, k, m** *$p < 0.05$, **$p < 0.01$ versus control (LV-GFP) group. Results were assessed for normality using the Shapiro–Wilk test and analysed using a two-tailed unpaired $t$ test. Data are shown as mean ± SEM ($n = 4$ per treatment). Detailed statistics and exact $p$ values are provided in Supplementary Table 8. Source data are provided as a Source Data file.

the injured spinal cord, suggesting an important role for CSPGs in the pathological chronification of the inflammatory response after SCI.

**CSPG digestion alters inflammatory gene co-expression dynamics during the resolution phase of the inflammatory response.** To generate an in-depth understanding of CSPG-mediated immunomodulatory effects, we performed a dynamic analysis of 29 target inflammatory-related genes (measured by qPCR on RNA extracted from the injury epicentre) at different post injury time points with or without CSPG digestion (study design, Fig. 2a). Multivariate pattern detection using dual multiple factor analysis (dMFA) revealed 2 main dynamic patterns (dimensions) over time, explaining ~56% of the total variation in gene co-expression (35.1% and 20.6% for dimension 1 and 2, respectively; Fig. 2b). Analysis of the correlated loadings (the importance of each cytokine within a dimension), reveals that dimension 1 is characterised by an overall positive correlation in cytokine expression, likely indicative of injury-induced activation of glial and immune cells, and dimension 2 by a positive correlation in Matrix Metalloproteinases (MMPs) and microglia/macrophage activation markers (CD68, CD206, CD163), likely indicative of tissue remodelling and immune cell activation (Fig. 2c and Extended Data Fig. 2). Comparing the scores of each group, we observed that both SCI groups significantly differed from naive animals at 6 h, 12 h, 1 dpi and 3 dpi. However, at 7 dpi whilst LV-GFP-treated animals were still significantly different from naïve animals, the profile of LV-ChABC-treated animals was restored to that of naive animals (Fig. 2d, Extended Data Fig. 3a and 3 dpi vs. 7 dpi—Extended Data Fig. 5g). Thus, CSPG digestion by LV-ChABC reduces pro-inflammatory cytokine gene expression to basal levels more quickly following injury. Bidimensional plots highlight the divergence between groups in the dimension 1 vs. dimension 2 space, where LV-ChABC animals cluster closely with naïve animals at 7 dpi (Fig. 2e and Extended Data Fig. 3b–d). We next sought to evaluate this at the cytokine profile level. Luminex multiplex immunoassay analysis revealed dynamic changes in cytokine expression levels from 6 h to 7 dpi, with several cytokines remaining elevated at 7 dpi (Extended Data Fig. 4). CSPG digestion reduced CXCL1 and IL6 at 1 dpi, which correlates with more rapid neutrophil clearance (resolution index: 52.8 vs. 40.3) (Extended Data Fig. 1d). However, little to no changes were observed at 6, 12 h or 3 dpi (Extended Data Fig. 4d–f, h and 3 dpi vs. 7 dpi—Extended Data Fig. 5f). A side by

side comparison of data at 3 dpi vs. 7 dpi revealed no significant differences in immune cell recruitment and phenotype, inflammatory cytokine protein synthesis or inflammatory gene expression extracted from dMFA after LV-ChABC treatment at 3 dpi (Extended Data Fig. 5a–g). In contrast, we observed significant changes in immune cell number and phenotype (Extended Data Fig. 5a–e), significantly reduced levels of pro-inflammatory cytokines CCL3, IL1α, CCL5, IL18, and IL1β (Extended Data Fig. 5f) and a significant reduction in numerous prototypical pro-inflammatory genes (iNOS, CD68, MHCII, IL1ß, TNFα, IL18, CCL2, CCL5, and CXCL10; Extended Data Fig. 5g) after lentiviral-ChABC treatment at 7 dpi. Thus, 7 dpi (which is a key time point in the resolution phase of inflammation[29]) was the earliest time-point when CSPG digestion elicited significant changes in multiple aspects of the inflammatory response after SCI, providing evidence that CSPG digestion attenuates pro-inflammatory cytokine release during the resolution phase of the inflammatory response.

Having found that CSPGs exhibit the most pronounced immunomodulatory effects at 7 dpi, we set out to further characterise this. To this end we investigated a larger number of cytokine genes at this time point. Principal component analysis (PCA) of the expression of 42 genes revealed 3 dimensions which account for ~73% of gene expression variance (40%, 20.4%, and 12.9% for dimension 1, 2, and 3, respectively; Fig. 2f). Dimension 1 was characterised by most cytokines moving in the same direction (positive correlated loadings). However, dimensions 2 and 3 were characterised by a patchwork of different cytokines moving in positive and negative directions (Fig. 2f). Comparing the scores for each dimension we observed that the LV-GFP-treated group significantly differed from the LV-ChABC-treated group in dimension 1, but not dimension 2 or 3 (Fig. 2g). This difference was characterised by LV-GFP moving in a positive direction and LV-ChABC moving in a negative direction on dimension 1 (Fig. 2g, h), confirming that at 7 dpi the expression profile of studied cytokines was strongly modulated by CSPG digestion. mRNA expression heatmaps (Fig. 2i) and univariate analysis of mRNA expression (Fig. 2j) show significantly reduced expression of cytokines and inflammatory molecules, assessed by qPCR, in animals treated with LV-ChABC compared to LV-GFP at 7 dpi. Together, these data show that modulation of CSPGs with ChABC has a significant impact on inflammatory gene dynamics and cytokine protein profile 7 days after SCI. In addition, for some of the inflammation-associated genes analysed, the observed reduction in inflammatory gene

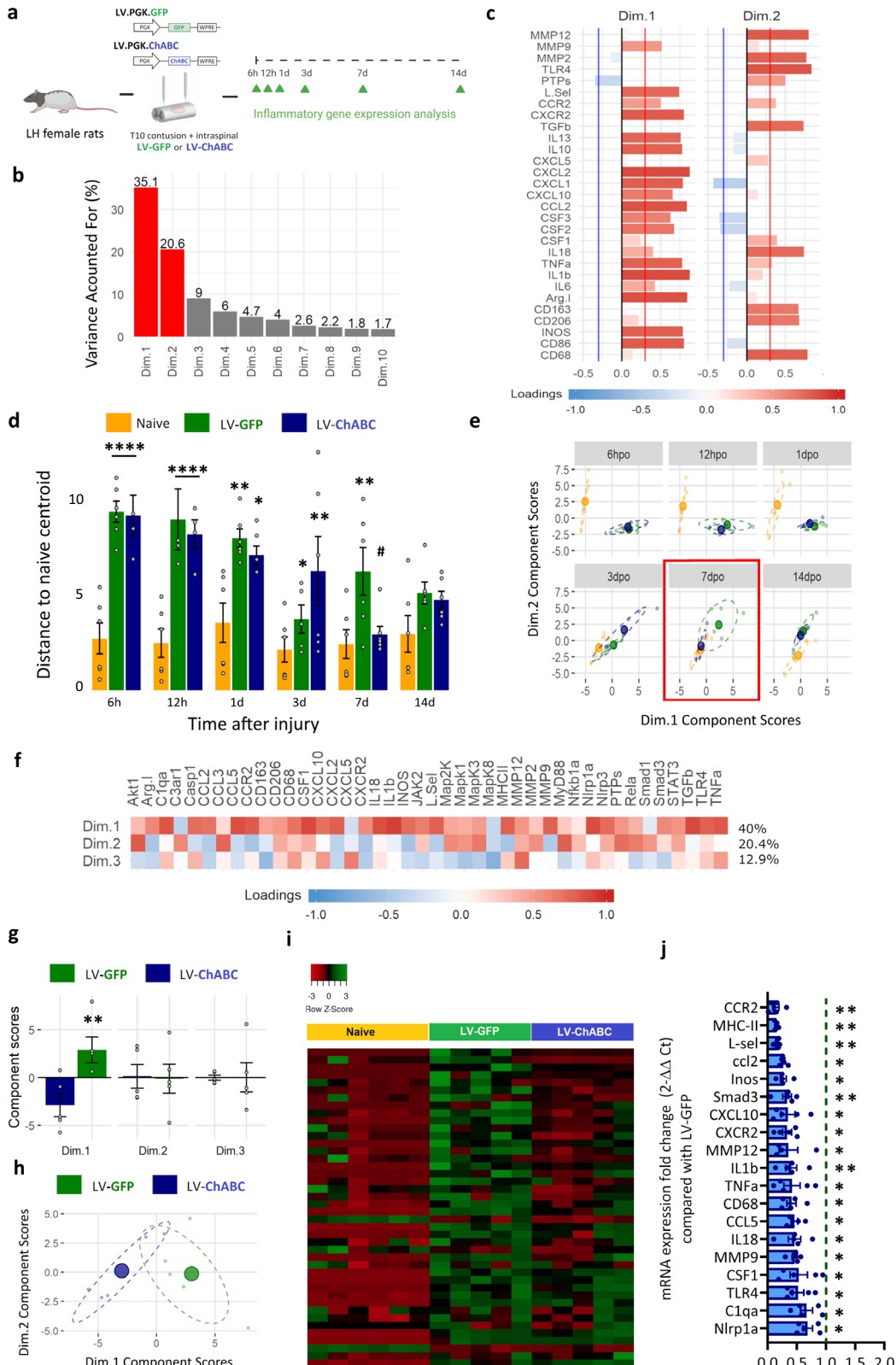

expression following CSPG digestion was maintained one week later (although to a lesser degree than at 7 dpi), alongside an observed increase in the pro-resolution marker CD163 (7 dpi vs. 14 dpi—Extended Data Fig. 6a). Thus, we demonstrate that the immunomodulatory effects of CSPGs after SCI are predominantly focused in the resolution phase of neuroinflammation where CSPGs modulate the inflammatory environment and obstruct the clearance of pro-inflammatory immune cells from the injury site.

## CSPGs modulate macrophage and microglial cell phenotype during the resolution phase of the inflammatory response.
Having probed gene expression at a whole tissue level, we next

**Fig. 2 CSPG digestion shifts inflammatory gene co-expression dynamics after spinal cord injury during the resolution phase of inflammation.**
**a** Experimental design of inflammatory gene co-expression analysis. **b**–**e** Dynamic analysis of the expression of 29 target genes over time with or without CSPG digestion. Gene expression was measured by qPCR on RNA extracted from the injury epicentre at different time-points after SCI. **b** Bar graph showing that multivariate pattern detection using dual multiple factor analysis (dMFA) detected 2 dynamic patterns (dimensions) over time explaining ~56% of the total variation in gene co-expression. **c** Loadings correlation bar graph to show which cytokines contribute to each of the patterns (interpreted as the Pearson $r$ correlation coefficient ranging from −1 to 1). **d** Bar graph representing the Euclidean distance to the naïve centroid for each group at different time-points, measuring the relative movement of treated animals with respect to naïve in the global inflammatory gene profile. Two-way ANOVA with time and group as factors using Tukey for multiple testing correction. $*p < 0.05$, $**p < 0.01$, $***p < 0.001$ versus uninjured (naive) group; $^{\#}p < 0.05$ versus LV-GFP group. Data are shown as mean ± SEM ($n$ = number of animals/samples, with $n = 6$ for each group–time combination except LV-GFP at 12 h, LV-ChABC at 6 and 12 h post injury ($n = 4$); LV-GFP at 3 dpi ($n = 5$) and LV-ChABC at 7 dpi ($n = 7$). **e** Bidimensional plots of the component scores in dimensions 1 and 2. Ellipsoids represent the bivariate standard deviation and the coloured circles the centroid. There is little divergence of LV-ChABC and LV-GFP at any timepoint except 7 dpi, where LV-ChABC becomes highly diverged from LV-GFP and is proximal to naïve, reflecting a gene expression pattern comparable to uninjured animals at 7 dpi after CSPG digestion. **f**–**i** Further pattern analysis at 7 dpi was performed using principal component analysis (PCA) for 42 inflammatory-related genes, confirming a reduction of pro-inflammatory genes after CSPG digestion, assessed by qPCR. **f** Loadings correlation heat map show dimension 1 loadings are positive for almost all cytokines, indicative of a global higher cytokine co-expression in LV-GFP vs. LV-ChABC-treated animals. **g** Component score bar graphs for each group and dimension at 7 dpi show significant differences between LV-GFP and LV-ChABC in dimension 1. $**p < 0.01$ versus control (LV-GFP) group. Results were assessed for normality using the Shapiro–Wilk test and analysed using a two-tailed unpaired $t$ test. Data are shown as mean ± SEM. (h) Bidimensional plot of the component scores for each group in dimension 1 and 2 at 7 dpi showing significant differences between LV-GFP and LV-ChABC in dimension 1. $**p < 0.01$ versus control group (LV-GFP) ($n = 5$ per treatment). **i** Heatmap showing gene expression data for 42 key genes in the inflammatory response at 7 dpi. LV-ChABC treatment elicits gene expression patterns closer to naïve than LV-GFP treated animals. **j** Bar graph showing all significant pro-inflammatory gene expression differences between LV-GFP and LV-ChABC treatments at 7 dpi. $*p < 0.05$, $**p < 0.01$ versus control (LV-GFP) group. Results were assessed for normality using the Shapiro–Wilk test and analysed using a two-tailed unpaired $t$ test. Data are shown as mean ± SEM ($n = 6$ naïve group, $n = 5$ per treatment). Detailed statistics and exact $p$ values are provided in Supplementary Table 8. Source data are provided as a Source Data file.

wanted to assess the phenotype of individual cell populations at 7 dpi. Using our t-SNE unbiased clusterisation based on lineage markers, we were able to identify microglia and monocyte/macrophages (Fig. 1f, g). We then sought to conduct an in-depth phenotype analysis of these populations. Firstly, in microglial cells (cluster 1, Fig. 1f–h) we assessed expression levels of particular markers linked to phenotype activation (M1-like: iNOS, CD68, CD86, MHC-II; M2-like: Arg I, CD206, and CD163; Fig. 3a–h). CSPG digestion significantly reduced the expression of MHC-II, a prototypical pro-inflammatory marker, in the microglial cell population (Fig. 3a, b). This MHC-II reduction at 7 dpi was maintained at least until 14 dpi (Extended Data Fig. 6b). These data indicate that CSPGs are a key mediator of inflammatory microglial activation after SCI. Next, we characterised the impact of CSPG digestion on the recruited monocyte/macrophage population, identified as cluster 2 and 3 by the high expression of CD45 and CD11b in our t-SNE unbiased clusterisation (Fig. 1f, g). Similar to microglial cells, infiltrated monocyte/macrophages from animals treated with LV-ChABC showed a significant reduction of MHC-II expression at 7 dpi (Fig. 3c, d), which was maintained at 14 dpi (Extended Data Fig. 6c). CSPG digestion also elicited additional immunomodulatory effects in monocytes/macrophages. Unlike in microglial cells, LV-ChABC treatment significantly reduced the expression of the pro-cytotoxic enzyme iNOS and the inflammatory activation marker CD68 in the monocyte/macrophage population (Fig. 3c, d). Although there appeared to be a trend for increased expression of anti-inflammatory M2-like markers with LV-ChABC treatment, this was not statistically significant (Fig. 3c).

The immunomodulatory effects of CSPG digestion on microglial and monocyte/macrophage immune cell populations at 7 dpi was further confirmed by phenotype gene expression analysis. Expression of pro-inflammatory and pro-repair-associated genes, assessed by qPCR on manual-gating microglia (GPR34$^{high}$, FcRIs$^{high}$, and CCR2$^{low}$) and monocyte/macrophage (GPR34$^{low}$, FcRIs$^{low}$, and CCR2$^{high}$) (Fig. 3i, j) sorted cells, revealed that CSPG digestion with LV-ChABC treatment redirects monocytes/macrophages and microglial cells toward a pro-repair (M2-like) phenotype after SCI (Fig. 3k). Changes in M1-like and M2-like

gene expression were no longer apparent by 14 dpi, other than reduced CD68 expression in microglia (Extended Data Fig. 6d, e). Finally, phenotype analysis was performed on the two monocyte/macrophage subsets identified in our t-SNE unbiased clusterisation (based on differential expression of CD43; Fig. 1l). This revealed that the immunomodulatory role of CSPGs in the total monocyte/macrophage population is predominantly mediated by their impact on the CD43$^{low}$ population, where CSPG digestion significantly reduced the expression of iNOS, CD68 and MHC-II (Fig. 3e–h).

Together these data demonstrate that CSPG digestion after SCI converts the phenotype of microglial cells and macrophages towards an anti-inflammatory, pro-repair state. This is particularly evident in the CD43$^{low}$ monocyte/macrophage population. Thus, CSPG digestion can activate inflammation resolution after SCI, thus overcoming the prolonged pro-inflammatory response that leads to detrimental/pathology and failure of tissue repair.

**Differences in immune cell spatial distribution after CSPG digestion.** We next sought to understand the immunomodulatory effects of CSPG digestion in-situ. To this end we used immunohistochemistry in spinal cord tissue sections to assess the expression and distribution of two prototypical markers for M1-like and M2-like cells following contusion SCI and treatment with either LV-GFP or LV-ChABC at 7 dpi. Expectedly, LV-ChABC treatment led to significantly less staining for intact CSPGs (CS-56 expression), in both perilesional and entire spinal cord areas (Fig. 4a, b, g, h). The spatial distribution of cells positive for the M2-like marker CD206 was dramatically altered by CSPG-digestion. In LV-GFP treated control animals, we observed CD206+ cells to be localised almost exclusively in a ring-like pattern within the inner astroglial border in close proximity to GFAP+ projections (Fig. 4cii), while CD206+ cells were largely absent from the injury core (Fig. 4ci). In contrast, while a similar pattern of CD206+ cells was observed along the astroglial border in close proximity to GFAP+ projections (Fig. 4dii), we also observed an abundance of CD206+ cells within the injury core following CSPG digestion (Fig. 4di). Interestingly, the densely

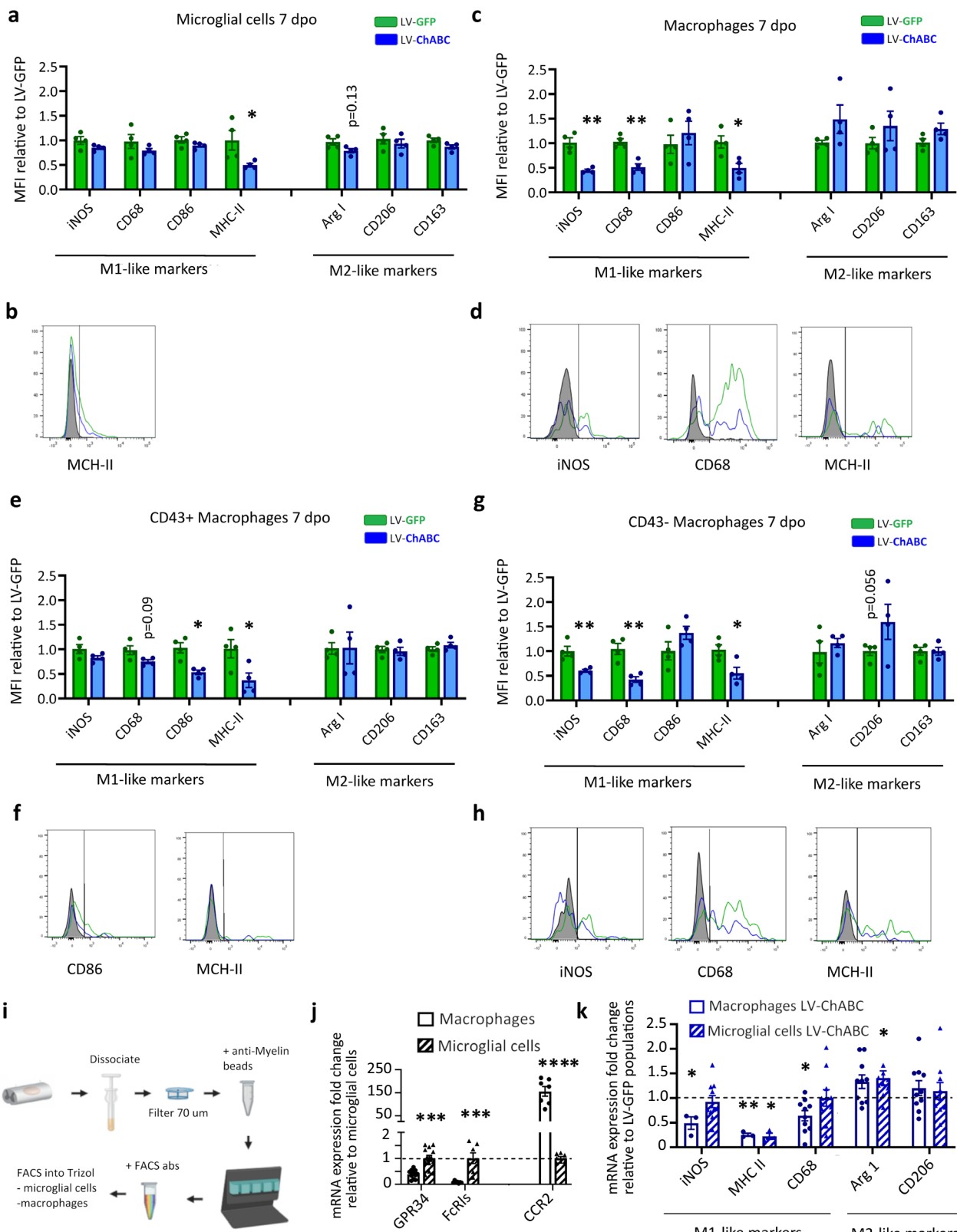

packed core of CD206+ immune cells was mirrored (almost exactly) by an absence of CSPGs, as can be seen by comparing Fig. 4a, c (dense CSPG in core, CD206+ cells excluded from the core) with Fig. 4b, d (absence of CSPG in core, abundance of CD206+ cells in the core). Increased CD206 expression in LV-ChABC treated animals was also mirrored by a reduction in the M1-like marker iNOS (Fig. 4e, f, j) within the lesion

core (Fig. 4fi.) and border (Fig. 4fii.), where iNOS expression appeared to be associated with areas of injury-induced loss of neurofilament (NFH). Therefore, these data confirm our molecular biology findings in-situ and provide spatial context to where CSPG–immune interactions occur. Notably, we provide evidence that CSPG deposition at the injury core restricts immune cells in this region from adopting a pro-repair phenotype.

**Fig. 3 CSPG digestion alters phenotypically distinct immune cell clusters during the resolution phase of inflammation after spinal cord injury. a** Bar graphs showing changes in the expression of classic M1 and M2 markers in the microglial cell population (cluster 1 in t-SNE analysis, Fig. 1f, g) at 7 dpi. Microglial cells exhibit significantly reduced expression of the pro-inflammatory (M1) marker MHC II in the LV-ChABC treated group, compared to LV-GFP treated animals. **b** FACS plot histogram of MHC-II expression in microglial cells at 7 dpi showing reduced expression in the LV-ChABC-treated group (blue) compared with control treatment (LV-GFP; green). Grey colour represents the isotype control. **c** Bar graphs showing changes in the expression of classic M1 and M2 markers in in the macrophage cell population (clusters 2 and 3 in t-SNE analysis, Fig. 1f, g), at 7 dpi. Macrophages exhibit significantly reduced expression of pro-inflammatory (M1) markers in LV-ChABC-treated animals compared to LV-GFP controls. **d** FACS plot histogram of M1 markers in the macrophage population at 7 dpi showing reduced expression in LV-ChABC treated group (blue) compared with control treatment (LV-GFP; green). Grey colour represents the isotype controls. **e, g** Graphs showing the changes in the expression of M1 and M2 markers in CD43$^{high}$ and CD43$^{low}$ macrophages (t-SNE cluster 3 and 2, respectively, Fig. 1f, g, i), at 7 dpi. Pro-inflammatory M1 marker reduction exerted by CSPG digestion is higher in the CD43$^{low}$ population (G). **f, h** FACS plot histograms of pro-inflammatory marker expression IN CD43$^{high}$ and CD43$^{low}$ macrophages, respectively, at 7 dpi showing reduced expression in the LV-ChABC treated group (blue) compared with control treatment (LV-GFP; green). Grey colour represents the isotype controls. **a, c, e, g** *p < 0.05, **p < 0.01 versus control (LV-GFP) group. Results were assessed for normality using the Shapiro–Wilk test and analysed using a two-tailed unpaired *t* test. Data are shown as mean ± SEM (**a** and **c**: LV-GFP n = 9, LV-ChABC n = 11; **e** and **g**: n = 4 per treatment). MFI mean fluorescence intensity. **i** Experimental design for phenotype gene expression analysis in sorted cells at 7 dpi. **j** Expression levels of microglial (GPR34 and FcRls) and monocyte/macrophage (CCR2) enriched genes evaluated by qPCR in sorted cells. ***p < 0.001, ****p < 0.0001 versus sorted microglial gene expression. Results were assessed for normality using the Shapiro–Wilk test and analysed using a two-tailed unpaired *t* test. Data are shown as mean ± SEM (n = 13 per cell population). **k** gene expression of classic M1 and M2 phenotype markers measured by qPCR in sorted monocytes/macrophages and microglial cells at 7 dpi. LV-ChABC treatment redirects monocytes/macrophages and microglial cells toward a pro-repair (M2) phenotype after SCI. *p < 0.05, **p < 0.01 versus normalised control group (LV-GFP treatment). Results were assessed for normality using the Shapiro–Wilk test and analysed using a two-tailed unpaired *t* test. Data are shown as mean ± SEM (iNOS, MHC-II n = 3; CD68, Arg I, CD206 n = 10 per treatment for macrophages; MHC-II n = 3; iNOS, CD68, Arg I, CD206 n = 10 per treatment for microglial cells). Detailed statistics and exact *p* values are provided in Supplementary Table 8. Source data are provided as a Source Data file.

**CSPG digestion reduces adaptive immune cell infiltration and modulates T$^{CD4}$ cell composition after SCI.** To gain a more comprehensive understanding of CSPG-immune modulatory effects, we next evaluated the role of CSPGs in adaptive immune cell recruitment, namely T-helper lymphocytes (TCD4), cytotoxic T lymphocytes (TCD8) and B cells. Using flow cytometry, we assessed the recruitment of TCD4 (CD45+$^{high}$, CD11b−, CD3+, CD4+ and CD8−), TCD8 (CD45+$^{high}$, CD11b−, CD3+, CD4−, and CD8+) and B cells (CD45+$^{high}$, CD11b−, CD3−, and CD45RA+) into the injury epicentre at 7, 14, and 28 dpi (Fig. 5a–d). Animals treated with LV-ChABC showed a significant reduction of CD4+ and CD8+ T cell recruitment at 7 dpi (Fig. 5b). A trend for reduced TCD4 numbers at later time points (14 and 28 dpi) was observed, although this did not reach statistical significance (Fig. 5c, d). Thus, CSPG digestion with LV-ChABC modulates the infiltration of T lymphocytes into the injury epicentre, indicating a role for CSPGs in modulating the adaptive immune response after SCI. TCD4 lymphocytes, also called T helper (Th), are a heterogeneous population which once activated can adopt a myriad of phenotypes depending on environmental signals[30]. We therefore investigated whether LV-ChABC treatment modulates the infiltration of specific Th subtypes. Th signature gene expression, assessed by qPCR in TCD4 sorted cells, showed that following LV-ChABC treatment there was a significant reduction of Th1-specific inflammatory gene expression at 7 dpi (Fig. 5e, f). We found that T-bet (a critical transcription factor for Th1 polarisation) and TNF-a (a prototypical Th1 cytokine) were significantly downregulated in sorted TCD4 lymphocytes after CSPG digestion (Fig. 5e). Furthermore, the expression of interferon gamma (IFNγ), a prototypical Th1 cytokine, was undetectable in TCD4 sorted cells from LV-ChABC-treated animals, in comparison to high expression in LV-GFP-treated animals (Fig. 5f). These data indicate that CSPGs are involved in the chronification of the inflammatory response after SCI by propagating the infiltration of TCD8 and TCD4 pro-inflammatory lymphocytes (Th1) at later stages of the inflammatory response.

Together, our in vivo data provides robust evidence that CSPG digestion can positively modulate both the innate and adaptive immune cell response after SCI, thereby enhancing multiple aspects of the resolution of inflammation.

**CSPGs promote phenotypic conversion of bone marrow-derived macrophages and isolated microglia from pro-resolving to pro-inflammatory.** In order to gain greater insight into the mechanisms by which CSPGs prolong inflammation, we next conducted cell culture experiments to assess the effect of CSPGs on specific immune cell-types. We first cultured bone marrow-derived macrophages (BMDMs) and polarised them towards M1-like (pro-inflammatory) and M2-like (anti-inflammatory) states to mimic their activation at early and resolution stages of inflammation, respectively (Fig. 6a). After evaluating purity (~98%), confirming polarisation (by flow cytometry and qPCR) and optimising stimulation parameters (Extended data Fig. 7), we investigated the effects of CSPG treatment (for either 4 or 16 h) on inflammatory gene expression in both macrophage phenotypes. We observed a slight reduction in inflammatory-related gene expression in M1-like polarised macrophages following 4 h CSPG stimulation, however these were small in magnitude (fold change <1; Fig. 6c, d). In contrast, M2-like polarised macrophages underwent dramatic phenotypic conversion following 4 h CSPG stimulation. CSPG treatment caused an almost complete reversal of M2-like phenotype towards a more pro-inflammatory phenotype, with significantly enhanced expression of multiple pro-inflammatory genes related with M1-like activation (Fig. 6c, e). Immunomodulatory effects of CSPGs were maintained at 16 h, although to a lesser extent than at 4 h, evident by PCA analysis of inflammatory gene expression profiles, where gene expression changes induced by CSPGs are most pronounced in M2-like compared to M1-like polarised macrophages, and at 4 h compared to 16 h (Fig. 6b). As with 4 h treatment, the predominant effects with 16 h CSPG treatment were observed in M2-like polarised macrophages, where CSPG treatment stimulated gene expression changes indicating conversion of M2-like macrophages to a more M1-like phenotype (Fig. 6f–h). Having observed differential effects of CSPG stimulation on macrophage gene expression, we next asked how CSPG stimulation affects macrophage function/behaviour. Phagocytosis is an important mechanism to recover tissue homoeostasis by which macrophages remove injury-induced cellular and environmental debris. In an assay which reflects the ability of macrophages to perform phagocytosis we observed that M2-like macrophages, but not M1-like macrophages, phagocytose significantly less when

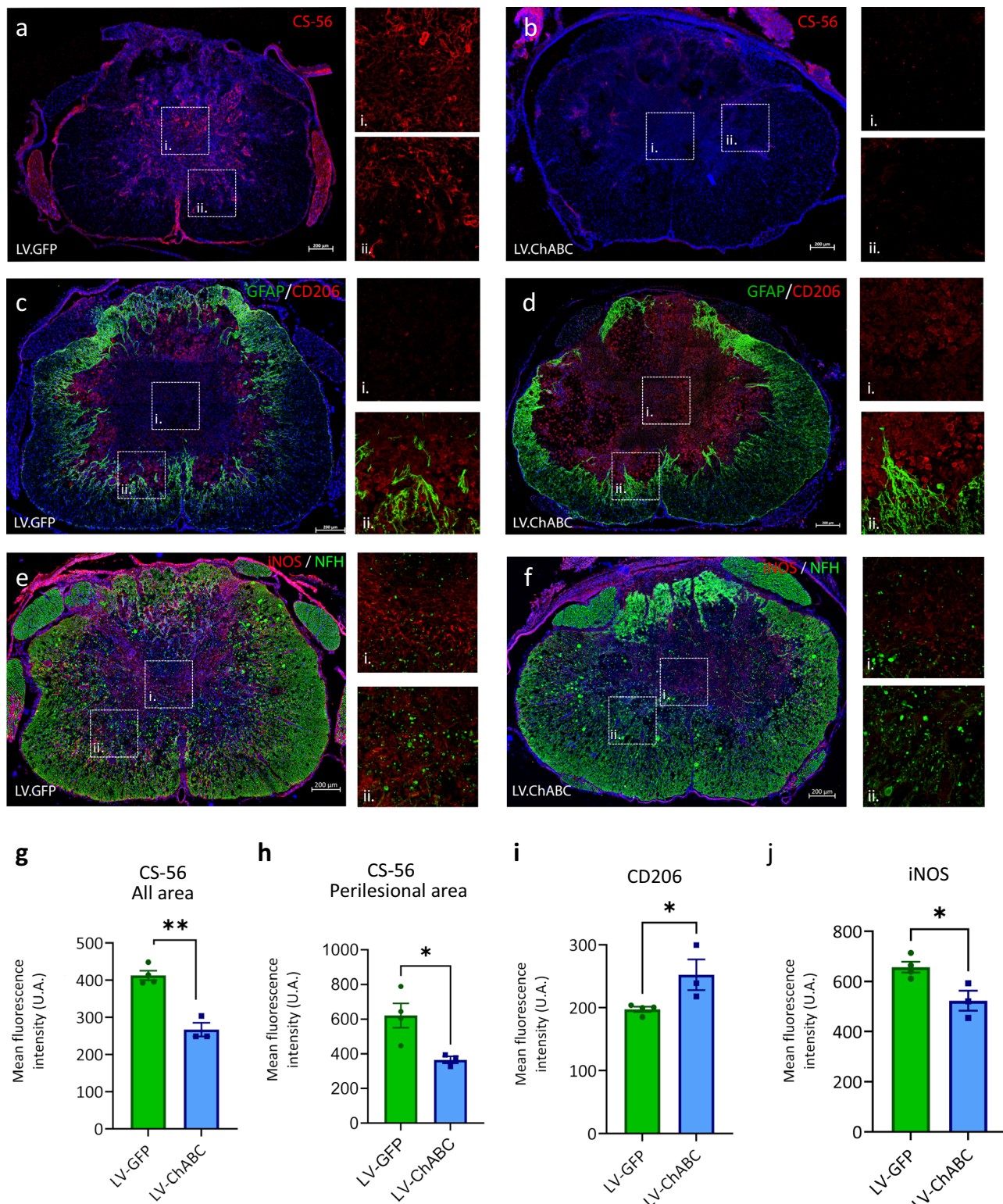

stimulated with CSPGs (Extended Data Fig. 8). Thus, at both a gene expression level and a functional level, CSPG stimulation makes M2-like macrophages more similar to M1-like macrophages. These data indicate that, in vitro, CSPGs have differential effects on macrophage phenotypes and a striking ability to convert pro-resolving macrophages to pro-inflammatory. Taken together with our in vivo findings, where positive immunomodulatory effects of CSPG digestion after SCI are prominent at the resolution phase, when it is important that macrophages undergo a phenotypic switch from pro-inflammatory to pro-repair, this provides evidence that CSPGs act directly on macrophages to prevent a switch to a reparative phenotype.

After evaluating the immunomodulatory effects of CSPGs in BMDMs, we next investigated CSPG-mediated immunomodulation in M2-like polarised isolated microglial cells. Primary microglial cell cultures were derived from P2-3 rat pups (Extended Data Fig. 9a) or adult rat brain and spinal cord (Extended Data Fig. 9b). After astrocyte and myelin removal and

**Fig. 4 CSPG digestion alters immune cell phenotype and spatial distribution at the injury epicentre after spinal cord injury. a, b** Immunohistochemistry in transverse spinal cord sections at the injury epicentre at 7 dpi showing CS-56 expression and distribution in **a** LV-GFP and **b** LV-ChABC treated animals (i and ii show higher magnification of selected areas). **c, d** Immunohistochemistry showing GFAP (green) and CD206 (red) expression and distribution in **c** LV-GFP and **d** Lv-ChABC treated animals (i and ii show higher magnification of selected areas). Note the change in spatial distribution of CD206+ immune cells in response to CSPGs, with CD206+ cells restricted to a ring-like pattern around the inner astroglial border and absent from CSPG-dense lesion core in LV-GFP-treated animals (**a, c**), in contrast to the densely packed core of CD206+ immune cells in LV-ChABC treated animals (**d**) mirrored almost exactly by an absence of CSPGs in the lesion core (**b**). **e, f** Immunohistochemistry showing NFH (green) and iNOS (red) expression and distribution in **c** LV-GFP and **d** LV-ChABC-treated animals (i and ii show higher magnification of selected areas). **a–f** Nuclei in blue were stained with DAPI. **g–j** Bar graphs quantifying CS-56 (**g, h**), CD206 (**i**) and iNOS (**j**) expression between groups assessed by fluorescence intensity. Results were assessed for normality using the Shapiro–Wilk test and analysed using a two-tailed unpaired $t$ test. $*p < 0.05$, $**p < 0.01$ vs. LV−. Data are shown as mean ± SEM ($n = 4$ in LV-GFP and $n = 3$ in LV-ChABC groups). Detailed statistics and exact $p$ values are provided in Supplementary Table 8. Source data are provided as a Source Data file.

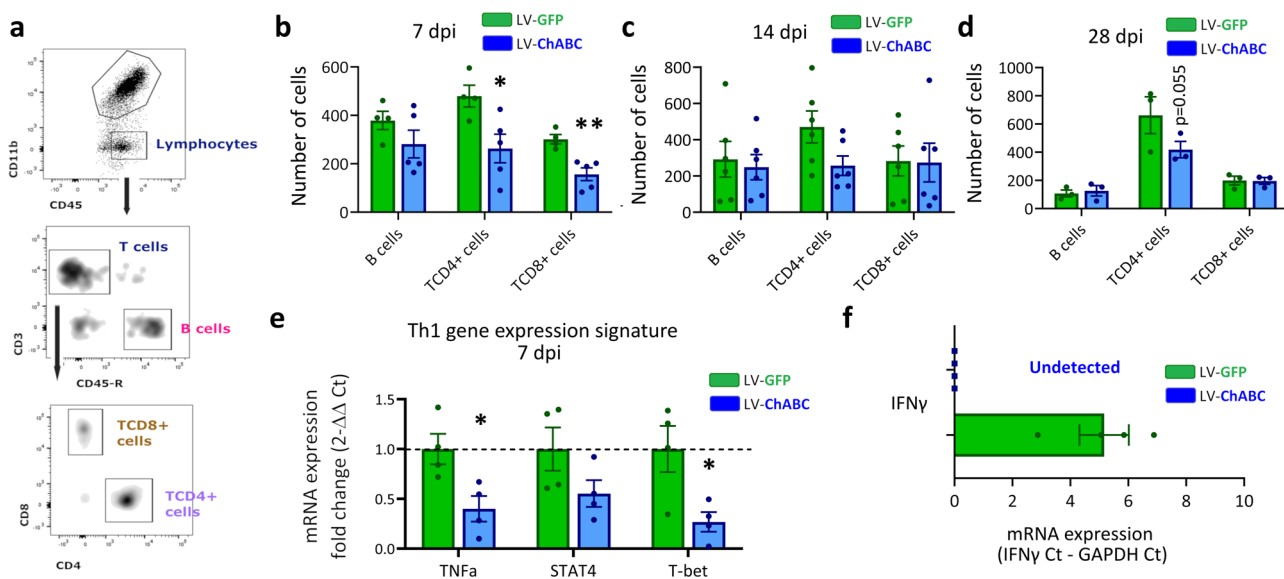

**Fig. 5 CSPG digestion reduces T cell infiltration and Th1 signature after spinal cord injury. a** Gating strategy used for lymphocyte recruitment assessment. **b–d** Graphs showing quantification of lymphocyte recruitment following SCI at **b** 7, **c** 14, and **d** 28 dpi with (LV-ChABC) or without (LV-GFP) CSPG digestion showing a significant reduction of T cell infiltration after CSPG digestion at 7 dpi. $*p < 0.05$, $**p < 0.01$ versus control (LV-GFP) group. Results were assessed for normality using the Shapiro–Wilk test and analysed using a two-tailed unpaired $t$ test. Data are shown as mean ± SEM ($n = 5$ at 7 dpi, $n = 6$ at 14 dpi, and $n = 3$ at 28 dpi per treatment). Data are pooled from at least two independent experiments. **e** Th1 signature gene expression assessed by qPCR in TCD4 sorted cells. **f** IFNg expression comparison between LV-GFP and LV-ChABC treated animals in TCD4 sorted cells at 7 dpi. **e, f** $*p < 0.05$ versus control (LV-GFP) group. Results were assessed for normality using the Shapiro–Wilk test and analysed using a two-tailed unpaired $t$ test. Data are shown as mean ± SEM ($n = 4$ per treatment). Detailed statistics and exact $p$ values are provided in Supplementary Table 8. Source data are provided as a Source Data file.

evaluation of purity, microglial cells were polarised towards an M2 anti-inflammatory state and gene expression analysed by qPCR (Extended Data Fig. 9a–c). We showed that CSPG treatment (4 h) significantly enhances the expression of pro-inflammatory cytokine genes (Il1β, iNOS, TNFα, and CCL3) in both adult and neonatal M2 polarised microglial cells (Extended Data Fig. 9d). In line with our in vivo findings (Fig. 3), we found that both adult and neonatal microglial cells were less sensitive to CSPG treatment than BMDMs, exhibiting lower up-regulation of inflammation-related genes (BMDM data from Fig. 6 combined for comparison in Extended Data Fig. 9). These data demonstrate that CSPGs exert a direct effect on inflammatory gene activation in both macrophages and microglial cells.

**Immunomodulatory effects of CSPGs in M2 macrophages are attenuated by ChABC in vitro and in vivo.** Having shown that CSPGs directly modulate macrophage and microglia phenotypes, we next set out to address whether CS-GAG removal using ChABC would affect this. BMDMs were transduced with LV-ChABC and the conditioned media was harvested and used as a

source of ChABC enzyme, with LV-GFP conditioned media as control (Fig. 7a). After evaluating the optimal vector titration (Fig. 7b–d), we assessed the effects of ChABC conditioned media on inflammatory cytokine expression in both M1-like and M2-like BMDMs (Fig. 7e, f). In M2-like BMDMs, incubation with ChABC conditioned media significantly attenuated the CSPG-mediated induction of IL1β, IL6, CCL5 and CXCL10 inflammatory gene expression (Fig. 7e). Thus, CSPG digestion with ChABC prevents M2-like BMDMs from phenotypic conversion upon exposure to CSPGs. Consistent with our previous observations (Fig. 6), M1-like polarised BMDMs did not upregulate IL1β, IL6, CCL5 and CXCL10 expression appreciably following CSPG stimulation (Fig. 7f). Interestingly, we observed that application of ChABC conditioned media reduced the basal levels of IL1β, IL6, CCL5, and CXCl10 in M1-like polarised BMDMs. This suggests that digestion products may positively modulate the inflammatory properties of M1-like macrophages. To this end, we asked whether CSPG digestion products themselves exert immunomodulatory effects. We evaluated the effect of the glycosaminoglycan digestion products (Chondroitin disaccharide Δdi-0S and Δdi-4S)

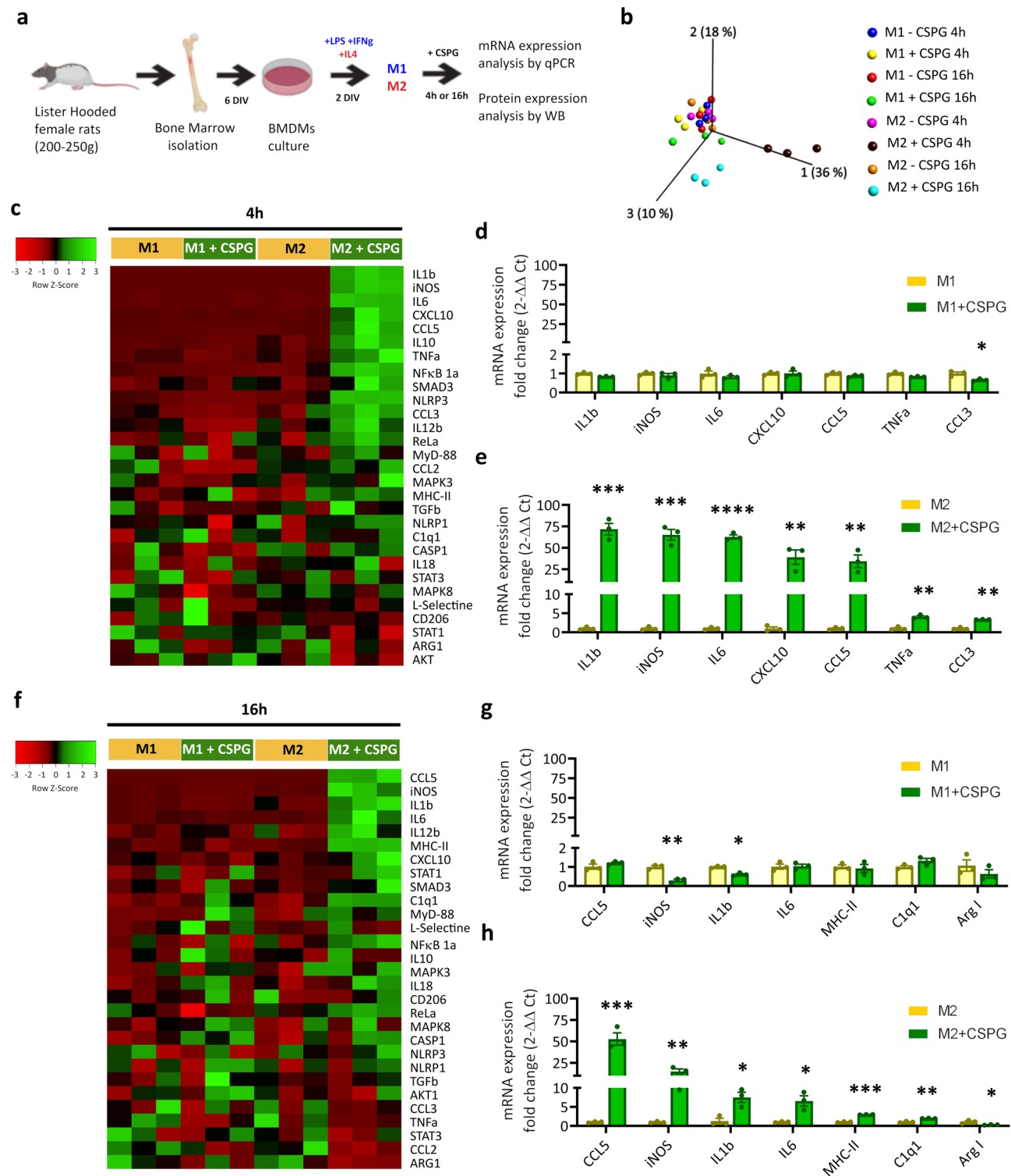

and compared with CSPGs on M1-like and M2-like polarised macrophages. Consistent with our previous experiments (Fig. 6) we observed that intact CSPGs cause a profound increase in pro-inflammatory gene expression in M2-like, but not M1-like macrophages (Extended data Fig. 10a, b). In contrast, the disaccharides did not elicit pro-inflammatory effects in either M1-like (Extended data Fig. 10a) or M2-like (Extended data Fig. 10b) polarised macrophages. One of the CSPG digestion products, the Chondroitin disaccharide Δdi-4S, was actually associated with higher expression of the M2-like signature marker CD206 in M2-

like macrophages, which could indicate an additional beneficial role for the digestion product itself in promoting an M2-like phenotype. Both macrophage and microglial responsiveness to CSPG digestion was next corroborated in vivo in FACS sorted microglial and macrophage cells from contusion injured spinal cords at 7 dpi (Fig. 7g; phenotype characteristics of sorted cells confirmed in Fig. 3j). Both macrophages and microglial cells exhibited reduced expression of inflammatory cytokine genes (Il1β, CCL5 and CXCL0) after CSPG digestion at 7 dpi, with the effect more pronounced in macrophages (Fig. 7h). Thus,

**Fig. 6 CSPG treatment converts anti-inflammatory macrophages to a pro-inflammatory phenotype. a** Experimental design of CSPG phenotype conversion studies in polarised bone marrow-derived macrophages (BMDMs) in vitro. **b** 3D PCA of inflammatory response gene expression profiles in M1 and M2 polarised BMDMs with or without CSPG treatment at 4 or 16 h ($n = 3$ per group). Note that gene expression alteration in BMDMs produced by CSPG treatment is highest in M2 polarised BMDMs compared to M1, and at 4 h compared to 16 h after the treatment. **c** Heatmap showing the effect of 4 h CSPG treatment (5 µg/ml) in M1 (left) and M2 (right) polarised BMDMs. CSPG immunomodulatory effects are predominant in M2 polarised BMDMs, causing a significant increase in multiple pro inflammatory genes. **d**, **e** Bar graphs showing genes that were significantly altered by 4 h CSPG treatment in **d** M1 and **e** M2 BMDMs. mRNA levels of inflammatory response genes were determined by qPCR. *$p < 0.05$, **$p < 0.01$, ***$p < 0.001$, ****$p < 0.0001$ vs. control (no CSPG). Results were assessed for normality using the Shapiro–Wilk test and analysed using a two-tailed unpaired $t$ test. Data are shown as mean ± SEM ($n = 3$ per group). **f** Heatmap showing the effect of 16 h CSPG treatment (5 µg/ml) in M1 (left) and M2 (right) polarised BMDMs. **g**, **h** Bar graphs showing genes that were significantly altered by CSPG treatment in (**g**) M1 and (**h**) M2 BMDMs. mRNA levels of inflammatory response genes were determined by qPCR. *$p < 0.05$, **$p < 0.01$, ***$p < 0.001$, ****$p < 0.0001$ versus control (no CSPG). Results were assessed for normality using the Shapiro–Wilk test and analysed using a two-tailed unpaired $t$ test. Data are shown as mean ± SEM ($n = 3$ per group). Detailed statistics and exact $p$ values are provided in Supplementary Table 8. Source data are provided as a Source Data file.

in vitro stimulation of polarised M2-like immune cells with CSPGs can elicit conversion from pro-repair to a pro-inflammatory phenotype. Conversely, degradation of CSPGs (either in CSPG-stimulated cells in vitro or in a pro-inflammatory SCI environment in vivo) can elicit a pro-repair immune cell phenotype. Taken together, these data indicate that CSPGs play a central role in phenotypic conversion of immune cells, such that in the presence of CSPGs macrophages are not able to adopt a pro-reparatory phenotype, but this can be enabled when CSPGs are degraded.

**The pro-inflammatory effect of CSPGs on M2 polarised macrophages is TLR4 dependent.** Having found that CSPGs directly cause M2-like polarised macrophages to adopt a proinflammatory M1-like phenotype, we asked how this occurs mechanistically. The TLR4 pathway has been recently implicated in proteoglycan signalling[31,32] and several of the proinflammatory cytokines that we consistently observed to be induced by CSPGs (IL1β, IL6, CCL5, CXCL10; Fig. 6 and Extended Data Fig. 10) are linked with TLR4 activation[33,34]. Moreover, TLR4 expression is highly linked with differences shown at later (resolution stage) time points (Extended data Fig. 2, dimensions 1 and 2). Therefore, we next determined the effects of CSPGs on macrophage phenotype when TLR4 signalling is inhibited, using either pharmacological TLR4 inhibition or genetic deletion of TLR4 (Fig. 8). In line with our previous results, CSPGs did not exhibit any effect on M1-like polarised macrophages (Fig. 8b). As expected, application of high (100 µM) and medium (10 µM) doses of a pharmacological inhibitor of TLR4 (TAK-242) had modest effects on pro-inflammatory gene expression in M1 polarised BMDMs with and without CSPG stimulation (Fig. 8b and Extended Data Fig 11c). This inflammatory gene expression reduction in M1-like BMDMs was not detected at TAK242 lower doses (2.5 µM; Extended Data Fig 11a). In contrast, while TAK-242 administration alone did not elicit any effects in gene expression in M2-like polarised macrophages, it completely abolished the induction of multiple proinflammatory cytokines (IL1β, iNOS, IL6, CXCL10, CCL2, CCL5, and TNFα) caused by exposure to CSPGs (Fig. 8c). These findings indicate that CSPGs act via the TLR4 pathway to induce proinflammatory activation in M2-like polarised macrophages.

We next asked whether other CSPG receptors (such as the well-described CSPG receptor PTPσ) may also contribute, or if they play a redundant role in CSPG-mediated immunomodulation. We therefore examined the effects of CSPGs on macrophage phenotype when PTPσ signalling is inhibited, using a PTPσ receptor inhibitor (intracellular sigma peptide, ISP; at concentrations previously described in the literature[26,35]: 2.5 and 10 µM), in a side by side dose comparison with TAK-242 (Extended Data Fig. 11). As previously described, the effects of TAK-242 in M1-

polarised BMDMs were modest, and we found that ISP also had negligible effects on reducing pro-inflammatory gene expression in M1-polarised BMDMs (Extended Data Fig. 11a, c). Interestingly, ISP elicited a small but significant increase in the M2-like marker CD206 in M1-polarised BMDMs at the lowest dose (Extended Data Fig. 11a), in agreement with a previous study[26]. We next assessed the two inhibitors in polarised M2-like macrophages (Extended Data Fig. 11b, d). A robust and significant reduction in pro-inflammatory gene expression in M2-like polarised macrophages was observed with TAK-242 at both doses. In contrast, PTPσ blockade had negligible effects in reducing CSPG-activated pro-inflammatory cytokine induction in M2-like polarised macrophages at either concentration. These findings reveal that CSPGs act predominantly via the TLR4 pathway, rather than via PTPσ, to cause M2-like polarised BMDMs to switch to a proinflammatory phenotype.

Having demonstrated the importance of TLR4 signalling in CSPG-mediated inflammatory macrophage activation, we asked whether TLR4 is differentially expressed in M1-like and M2-like macrophages. We found that M2-like polarised macrophages express more TLR4 than M1-like macrophages at a gene expression level (qPCR—Extended Data Fig. 12a) and protein level (flow cytometry—Extended Data Fig. 12b, c) in rat BMDMs. This result was replicated in mouse BMDMs, where TLR4 gene expression was significantly higher in M2- than M1-like polarised macrophages (qPCR—Extended Data Fig. 12d). Protein expression in mouse BMDMs was assessed by immunocytochemistry (Extended Data Fig. 12e, f), which showed higher constitutive expression of TLR4 in M2- than M1-like BMDMs. LPS is a potent TLR4 agonist that leads to the activation of the LPS/CD14/ TLR4 complex which promotes its subsequent endocytosis by endosomes[36]. Consistent with this, we found that TLR4 detection is higher in M1-like macrophages after cell permeabilization (indicating TLR4 internalisation), although it was still significantly lower than TLR4 expression in M2-like macrophages (Extended Data Fig. 12e, f). Finally, we assessed whether differential TLR4 expression could underly the differential effects of CSPGs on CD43− and CD43+ monocyte/macrophages at 7 dpi (Extended Data Fig. 12g–j). FACS-sorted CD43− monocyte/ macrophages showed significant higher TLR4 expression than its CD43+ counterpart assessed by flow cytometry (Extended Data Fig. 11g–i) and gene expression (qPCR—Extended Data Fig. 12j), suggesting that differential TLR4 expression likely contributes to the different immunomodulatory effects of CSPG on these populations.

In order to gain greater mechanistic insight into CSPG-mediated immunomodulation we evaluated MAPK and NF-kB pathway activation (the main pathways activated by TLR4) in rat M1-like and M2-like macrophages at 4 and 16 h (Extended Data Fig. 13a–d). CSPG treatment significantly upregulated the

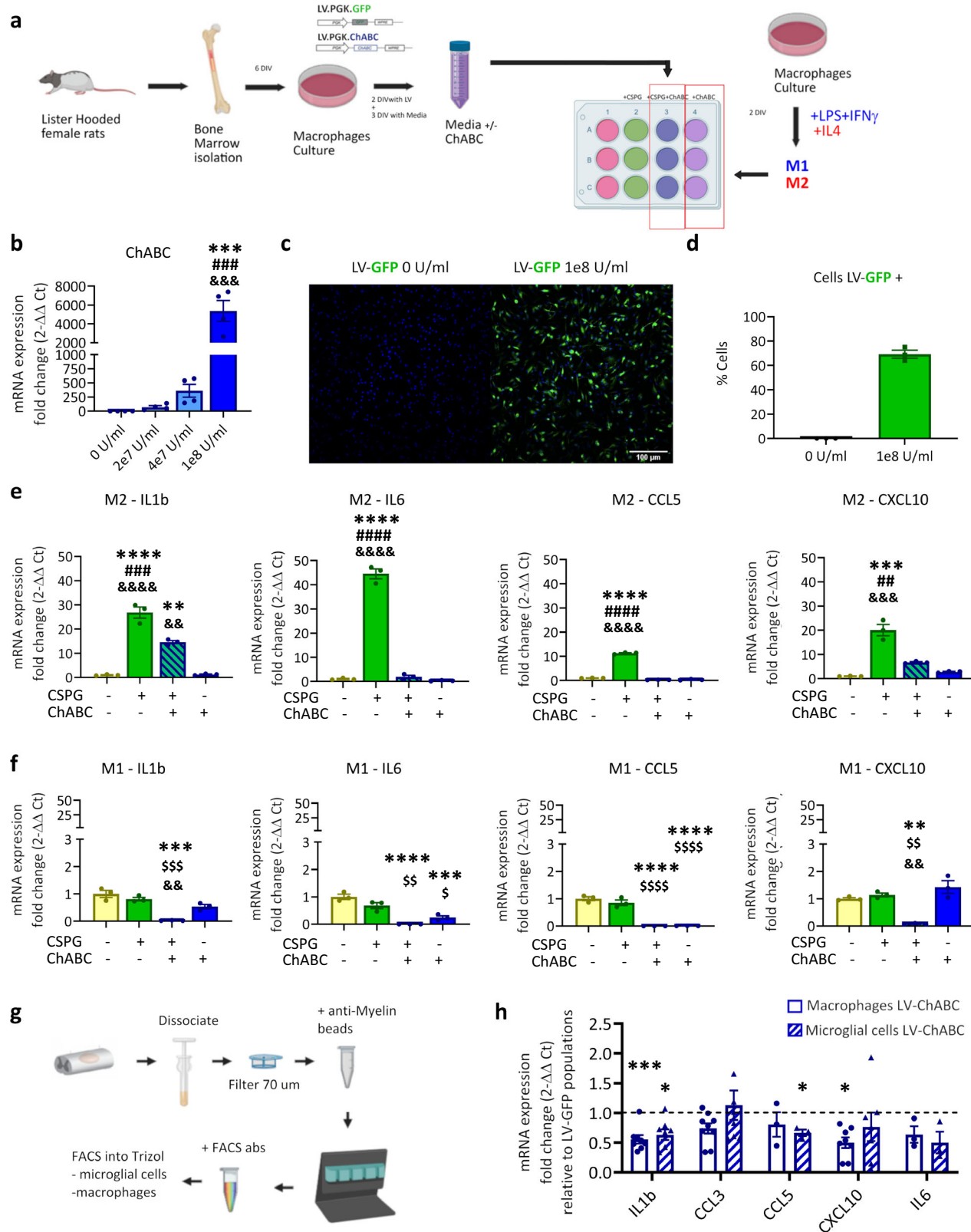

phosphorylation of p38 in M2-like polarised macrophages at both 4 and 16 h but did not affect other MAPK or NF-kB signalling pathways (Extended Data Fig. 13b, d). The effect of CSPGs on p38 pathway activation was corroborated by fluorescence intensity in mouse M1/M2-like polarised macrophages (Extended Data Fig. 13e, f) where 4 h CSPG incubation induced higher p-p38/p38 ratio fluorescence

intensity in M2-polarised macrophages compared with controls. The effect of CSPG on p38 pathway activation in M2-polarised macrophages was further corroborated in rat BMDMs, where p38 pathway inhibition using SB202190 (25 μM) significantly reduced the expression of several inflammatory cytokines (IL1b, iNOS, IL6, CXCL10, and CCL5) upregulated by CSPGs in M2-like macrophages. These data suggests that activation of the p38

**Fig. 7 CSPG digestion reduces the inflammatory effects of CSPGs in M2 polarised BMDMs in vitro and in vivo. a** Experimental design for evaluating the effects of ChABC conditioned media on polarised BMDMs after CSPG (5 μg/ml) activation. **b** mRNA level of ChABC gene expression produced by non-polarised (M0) BMDMs at different LV-ChABC titre transfection. Results were assessed for normality using the Shapiro–Wilk test and one-way ANOVA with Tukey post hoc test was used to analyse significant differences. ***$p < 0.001$ vs. control group (0 GC/ml), ###$p < 0.001$ vs. 2e7 GC/ml group, &&&$p < 0.001$ vs. 4e7 GC/ml group. Relative fold changes presented as mean ± SEM ($n = 3$ per group). The optimum titration was 1e8 GC/ml, which was used for further experiments. **c** Immunocytochemistry of BMDMs transfected with LV-GFP (green) to confirm (**d**) the percentage of transfection (68%) at 1e8 U/ml ($n = 3$ per group). Data are shown as mean ± SEM. Bar graphs of inflammatory gene expression by CSPG treatment with or without ChABC enriched medium in **e** M2 and **f** M1 BMDMs. mRNA levels of inflammatory response genes were determined by qPCR. **$p < 0.01$, ***$p < 0.001$, ****$p < 0.0001$ vs. control (no CSPG) group; #$p < 0.05$, ##$p < 0.01$, ###$p < 0.001$ vs. +CSPG+ ChABC group; &$p < 0.05$, &&$p < 0.01$, &&&$p < 0.001$, &&&&$p < 0.0001$ vs. +ChABC group; \$$p < 0.05$, \$\$$p < 0.01$, \$\$\$$p < 0.001$, \$\$\$\$$p < 0.0001$ vs. +CSPG group. Results were assessed for normality using the Shapiro–Wilk test and one-way ANOVA with Tukey post hoc test was used to analyse differences between conditions. Data are shown as mean ± SEM ($n = 4$ per group). **g** Experimental design for cytokine gene expression analysis in sorted cells from contused rat spinal cord at 7 dpi. **h** Bar graphs showing inflammatory cytokine gene expression measured by qPCR in sorted macrophages and microglial cells at 7 dpi, showing reduced pro inflammatory cytokine gene expression in response to CSPG digestion in both populations after SCI. *$p < 0.05$, ***$p < 0.001$ vs. normalised control group (LV-GFP treatment). Results were assessed for normality using the Shapiro–Wilk test and analysed using a two-tailed unpaired $t$ test. Data are shown as mean ± SEM (CCL5, IL6 $n = 3$; Il1b, CCL3, CXCL10 $n = 9$, per treatment and cell population). Detailed statistics and exact $p$ values are provided in Supplementary Table 8. Source data are provided as a Source Data file.

pathway plays a role in mediating CSPG-induced proinflammatory cytokine gene expression.

We next examined the involvement of TLR4 in phenotypic conversion of macrophages in greater detail, using BMDMs derived from a TLR4 knockout mouse (BMDMs$^{TLR4-/-}$), treated with the polarising factors used to establish M1-like and M2-like phenotypes in rat BMDMs (Fig. 8). As expected (due to the lack of TLR4 and in line with results obtained in Fig. 8b with TLR4 pathway inhibition), the pro-inflammatory M1-like activation was reduced, with M1-stimulated BMDMs$^{TLR4-/-}$ exhibiting an intermediate inflammatory phenotype with more amoeboid shape, reduced iNOS expression and lower basal inflammatory cytokine levels than their WT counterparts (Supplementary Table 6, Extended Data Fig. 14a–c). Furthermore, M1-like BMDMs$^{TLR4-/-}$ exhibited significantly higher CD206 expression, a well-known anti-inflammatory signature marker, than WT M1-like polarised macrophages, showing an intermediate inflammatory phenotype (Supplementary Table 6, Extended Data Fig. 14d). However, M2-like BMDM$^{TLR4-/-}$ activation was not compromised by the lack of TLR4, with both BMDM$^{TLR4-/-}$ and BMDM$^{WT}$ exhibiting typical M2 morphology and basal inflammatory gene expression levels (Supplementary Table 6, Extended Data Fig. 14a–c, e). We then explored the response of polarised BMDM$^{TLR4-/-}$ to CSPG stimulation. Unlike BMDM$^{WT}$ M0, which become activated by CSPG stimulation and upregulate proinflammatory cytokines CXCL10, IL1ß and TNFα, BMDM$^{TLR4-/-}$ M0 were largely unresponsive to CSPG stimulation (Extended data Fig. 14f). This suggests that TLR4 is critical for CSPG activation of M0 macrophages towards a proinflammatory status. In M1-stimulated BMDM$^{TLR4-/-}$, due to their ineffective inflammatory activation, there was a significant upregulation of IL1ß, CXCL10, and CCL2 (Fig. 8f and Extended data Fig. 14d). This contrasts with fully M1-activated BMDM$^{WT}$ which did not undergo appreciable changes in inflammatory gene expression (Fig. 8e and Extended data Fig. 14d; and in line with rat BMDM data, Fig. 6d, g). This suggests that in M1 macrophages that are only partially activated, CSPGs elicit proinflammatory effects via a TLR4-independent mechanism. In contrast, TLR4 appears essential for inflammatory activation of M2 macrophages, demonstrated by the significant upregulation of a number of prototypical proinflammatory cytokines (IL1ß, iNOS, IL6, CXCL10, CCL2, CCL5, and TNFα) upon CSPG stimulation in M2-stimulated BMDM$^{WT}$, whereas M2-stimulated BMDM$^{TLR4-/-}$ only upregulated CXCL10 (Fig. 8h and Extended data Fig. 14e). Finally, the intermediate polarisation exhibited by M1-like BMDM$^{TLR4-/-}$ (Extended Data Fig. 14d, Supplementary Table 6), namely the upregulation of IL1ß, CXCL10

in M1 BMDM$^{TLR4-/-}$ and the upregulation of CXCL10 in M2-like BMDM$^{TLR4-/-}$ may be partly explained by compensatory effects of other TLRs. Accordingly, we observed an increase in TLR2 in M1-like BMDM$^{TLR4-/-}$ and an increase in TLR2 and TLR6 in M2-like BMDM$^{TLR4-/}$ (Extended data Fig. 14g, h). Thus, using two approaches to inhibit TLR4 signalling in rat and mouse models we demonstrate that TLR4 is essential for the phenotypic switch that converts polarised pro-repair macrophages to a pro-inflammatory phenotype in the presence of CSPGs.

We have demonstrated, using multiple methods and using both genetic and pharmacological approaches, that CSPGs affect immunomodulation via TLR4 signalling. As CSPGs have well described effects on neuronal growth inhibition[9,37–39], we finally assessed whether CSPG interactions with TLR4 have any effect on neurite outgrowth. CSPG treatment caused a significant reduction in neurite length in both WT and TLR4 KO neurons, and this effect was rescued by application of ChABC equally in both groups (Extended Data Fig. 15). These results suggest that neuronal growth inhibition by CSPGs is not influenced by TLR4 signalling and that the main role of CSPG–TLR4 interactions is modulation of the immune cell response to injury.

Together these data reveal that CSPGs act via TLR4 to provoke a switch to a pro-inflammatory phenotype in pro-repair immune cells. This provides a new mechanism underlying the delayed resolution phase of inflammation following SCI, which leads to chronic pathology.

## Discussion

CSPGs are well established to be potent inhibitors of axonal growth and neuroplasticity after SCI. Here we report a role for CSPGs that goes far beyond growth inhibition and demonstrate their critical function as pro-inflammatory mediators that prevent pro-repair phenotypic conversion of immune cells during the resolution stage of inflammation. Finally, we reveal that these immunomodulatory effects are driven by TLR4.

Neuroinflammation in response to injury or disease is critical for enabling wound healing and tissue repair. However, active termination of neuroinflammation is required to successfully restore tissue homoeostasis. Failure of inflammation resolution can lead to impaired wound healing, chronic pathology and neurodegeneration, which are typical pathological hallmarks of SCI in humans[40] and in rodent models[41]. CSPGs have previously been implicated in neuroinflammation, although evidence as to their role is conflicting[26,42]. By conducting a dynamic and in-depth characterisation of the immunomodulatory role of CSPGs

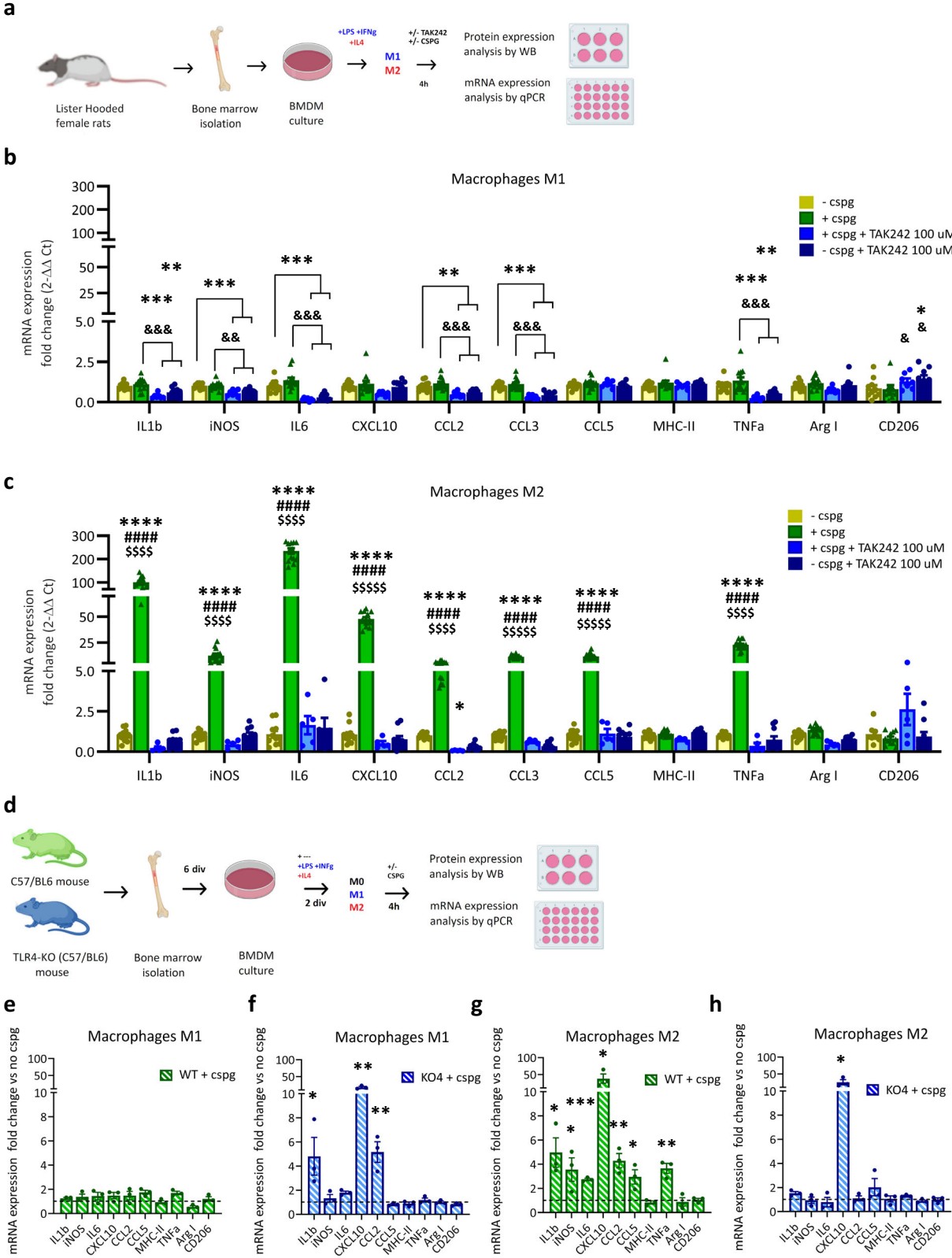

following SCI we delineate their effects on multiple cell types and cell phenotypes. We show that over-expression of the CSPG-digesting enzyme ChABC promotes enhanced clearance/reduced recruitment of immune cells, causes pro-resolution phenotypic changes in both innate and adaptive immune cells, and elicits dynamic immune signature changes in the tissue microenvironment at a key resolution time point.

Our data revealed that CSPGs are regulators of multiple cells types involved in innate and adaptive immune responses to SCI. The spinal injury epicentre is a complex milleu of cell types, activation states and phenotypes[2,23]. Whilst CSPGs have previously been implicated in signalling with resident glia and innate immune cells[2,3,43], until now we had little understanding of this effect on precise populations, or their phenotypes. Furthermore,

**Fig. 8 The TLR4 pathway is essential for the CSPG-activated inflammatory phenotype in M2 polarised macrophages. a** Experimental design to study the effect of CSPG treatment (5 µg/ml) on polarised BMDMs treated with or without TAK242 (100 µM), a pharmacological inhibitor of TLR4. Bar graphs showing the expression of inflammatory response genes in **b** M1 BMDMs and **c** M2 BMDMs. mRNA levels were determined by qPCR. Data were normalised with respect to control (no CSPGs). Results were assessed for normality using the Shapiro–Wilk test and one-way ANOVA with Tukey post hoc test was used to analyse differences between conditions. *$p < 0.05$, **$p < 0.01$, ***$p < 0.001$ vs. control group, $^{\&}p < 0.05$, $^{\&\&}p < 0.01$, $^{\&\&\&}p < 0.001$ vs. CSPG treated group, $^{\#}p < 0.05$, $^{\#\#}p < 0.01$, $^{\#\#\#}p < 0.001$ vs. +CSPGs+TAK242 group, $^{\$}p < 0.01$, $^{\$\$}p < 0.001$, $^{\$\$\$}p < 0.001$ vs. +TAK242 group. Data are presented as mean ± SEM ($n = 13$ in −CSPG and +CSPG groups; $n = 7$ in +CSPG +TAK242 group; $n = 10$ in −CSPG +TAK242 group). **d** Experimental design to compare the effect of CSPG treatment on BMDMs from C57/B6 WT and TLR4$^{-/-}$ mice. **e–h** Bar graphs comparing immunomodulatory effects of CSPG treatment (4 h at 5ug/ml) between WT and TLR4$^{-/-}$ polarised BMDMs. Differences in inflammatory gene expression by CSPG treatment were assessed in **e** M1− WT, (f) M1− TLR4$^{-/-}$, **g** M2-WT and **h** M2-TLR4$^{-/-}$ BMDMs. mRNA levels of inflammatory response genes were determined by qPCR. Data was normalised by their respective controls (WT or TLR4$^{-/-}$ BMDM without CSPGs), represented by dotted line. Results were assessed for normality using the Shapiro–Wilk test and analysed using a two-tailed unpaired $t$ test. *$p < 0.05$, **$p < 0.01$, ***$p < 0.001$ vs. WT w/o CSPGs. Data are shown as mean ± SEM ($n = 3$ per group). Detailed statistics and exact $p$ values are provided in Supplementary Table 8. Source data are provided as a Source Data file.

whether the adaptive immune response is modified by CSPG digestion has never been explored. To capture the complexity of cell phenotypes, we used 15 colour flow cytometry and multiple lineage markers to evaluate the role of CSPGs on recruitment and phenotype of different cell populations. We observed that removal of CS-GAGs significantly reduced the number of monocyte/macrophages present in the spinal cord at day 7 following spinal contusion injury. This is consistent with studies in inflammatory demyelinating disorders, where CSPG deposition has been shown to increase leucocyte migration[12,14] and treatment with Surfen, a CSPG- and HSPG-binding agent, can reduce macrophage recruitment into the CNS[13]. Furthermore, at a more granular level we were able to identify a differential effect on two monocyte subsets, classical CD43$^{low}$ monocyte/macrophages and non-classical CD43$^{high}$ populations, which are related with inflammatory and tissue repair functions, respectively[44]. CSPG digestion reduced proinflammatory classical CD43$^{low}$ recruitment, and marker profiling revealed a striking decrease in iNOS, CD68 and MHC class II expression in this population. Thus, CSPGs can directly influence the pro-inflammatory profile of specific monocyte subsets, placing them in a key position to modulate secondary injury inflammatory pathology.

Correspondingly we also found a significant reduction in the number of microglia following CSPG digestion at this timepoint. One prerequisite for resolution of inflammation is a switch in responding cells to a reparative phenotype. Macrophages and microglia sit on a broad spectrum of activation states, and here we used the classical M1 (pro-inflammatory) and M2 (anti-inflammatory) phenotype classification[45,46]. Increasingly precise characterisation of phenotypes is afforded by single-cell and high-plex-omics approaches[47] and these techniques are beginning to add greater clarity to which genes are true indicators of multiple diverse activation states in vivo. However, despite the caveat that macrophages/microglial cells will adopt multiple intermediate states, the notion of a spectrum of activation based on pro- vs. anti-inflammatory identities still represents a useful framework in which to position global phenotype. Despite simplification, M1-like and M2-like classification is still an effective, accepted method used to probe responses to injury and tissue remodelling[48,49]. Previous studies have shown that the SCI injury microenvironment inhibits M2-like polarisation[46,50]. In our own previous work, we found that CSPGs propagate secondary injury inflammatory pathology after SCI[24,25]. We now show that CSPGs are key mediators in the prevention of M2-like polarisation in both microglia and macrophages in vivo. In depth phenotype analysis in microglia showed that CSPGs modulate microglial activation, since CSPG digestion significantly reduced the expression of the prototypical proinflammatory marker MHC-II. In the context of recent work, highlighting the importance of

microglia in orchestrating wound healing[51–53], this places CSPGs as key players in the response to injury.

Until now the effects of CSPG digestion on the adaptive immune system have not been studied. We found that CSPG removal leads to an overall decrease in lymphocyte recruitment to the lesion. T-lymphocyte recruitment progressively increases within the first week post injury[54] and contributes to SCI pathology, by direct effects on neurons or glia or indirect effects on other CNS cells following the production of proinflammatory cytokines or chemokines[55]. There is evidence to suggest both positive and negative effects of T-lymphocyte recruitment following SCI[56,57]. We therefore sought to distinguish between T-lymphocyte subtypes in our analysis. Here we found that CSPG digestion results in significant reduction of CD4+ and CD8+ T cell recruitment at day 7 after contusion SCI. Furthermore, we found that CSPG digestion reduced type 1 T-helper (Th1) cells specifically. Within this population, CSPG digestion rendered IFNγ expression undetectable. IFNγ is a prototypical Th1 cytokine and thought to be an important mediator of cross-talk between T-lymphocytes and macrophages[58–60]. Our findings are consistent with previous work showing a modulation of Th1/Treg balance towards less inflammatory after inhibition of CSPG signalling via LAR and PTPσ receptors in a compressive SCI model[26] and reduced CD4+ T cell infiltration after surfen treatment in a murine EAE model[13]. Thus, CSPG targeting dampens the T-lymphocyte response. Whether CSPGs activate T-lymphocytes directly, or whether this is an indirect effect of the pro inflammatory microenvironment stimulated by CSPGs is the subject of further study. Collectively, these data demonstrate that CSPGs elongate the period of inflammation by modulating both the innate and adaptive immune response following SCI.

We further revealed that CSPG deposition at the injury epicentre restricts pro-repair immune cells from the lesion core. There is increasing appreciation as to the roles of specific cell types in the injured spinal cord and how their spatial distribution influences interactions[51–53,61]. Significant recent findings place activated microglial cells at a critical position at the interface between infiltrating leucocytes and border astrocytes, where they are associated with protection of neural tissue[53]. Similar to these findings, here we observed a ring-like pattern of CD206+ cells along the inner astroglial border after SCI. Additionally, we observed that CSPG digestion changes this restricted distribution and results in a dramatic increase in CD206+ cells. Thus, the spatial distribution of CSPG-immune interactions could play an important role in recently identified wound healing processes, such as the microglial scar[53] and/or the process of "corralling", where phagocytic immune cells are confined to the injury core which is surrounded by an astrocytic border[51]. In support of this, we found that CSPG digestion dramatically altered the

balance of pro- vs. anti-inflammatory immune markers within the lesion core. Notably, when CSPGs were degraded the core was no longer filled with toxic immune mediators such as iNOS but was abundant in pro-repair CD206+ immune cells (the densely packed core of CD206+ immune cells mirrored almost exactly by an absence of CSPGs). We therefore hypothesise that CSPGs prevent immune cells at the very injury core from converting to a pro-repair phenotype, and this leads to tissue necrosis at the injury epicentre and subsequent tissue cavitation. Thus, confinement of phagocytic immune cells may be beneficial early on[51,53], but then phenotype conversion is needed for inflammation resolution. If this does not occur, toxic macrophages at the core lead to tissue necrosis and central cavitation, which are classic pathological hallmarks of non-resolving SCI pathology[62,63]. We demonstrate that CSPGs play a critical role in this process.

Our next key finding demonstrated that CSPGs prevent resolution of inflammation following spinal cord injury. Although there is increasing evidence for a role of CSPGs in modulating inflammation following SCI[24–27], until now it has remained unclear how and at which stage of neuroinflammation this occurs. Our phenotype analysis of specific immune cell subpopulations revealed CSPG immunomodulatory effects to be most prominent at day 7 following spinal contusion injury, which is considered a key resolution time point when inflammatory mediators and immune cells need to switch towards a repairing phenotype[29]. Dynamic analysis of high-dimensional inflammatory gene expression and luminex measurements of cytokine levels revealed that at this critical time point, the presence of CSPGs results in a complex proinflammatory profile at a whole tissue level, captured by multiparametric analysis. Digestion of CSPGs using lentiviral overexpression of ChABC was found to accelerate resolution of this profile towards homoeostasis. This suggests that CSPG immunomodulatory effects predominate in the resolution phase of the inflammatory response.

Inflammation resolution, once believed to be a passive process involving downregulation of pro-inflammatory chemokine gradients, is an active anti-inflammatory programme involving specific pro-resolving mediators aimed at restoring tissue homoeostasis[19,64–66]. As outlined above, successful inflammation resolution requires activated pro-inflammatory immune cells to convert towards a more reparative phenotype[46,50]. After SCI, microglia and macrophages display a predominant and sustained pro-inflammatory/M1-like phenotype, which is thought to mediate cytotoxic actions and lead to excessive tissue damage. There is currently limited information about the in vivo factors that prevent these immune cells from converting to an M2-like phenotype, which is associated with tissue healing and repair. Recent work has suggested that CSPGs may influence macrophage polarisation[26], and this may underlie the neuroprotective effects observed after large scale CSPG digestion[24]. Here we have provided direct in vivo evidence that CSPGs are involved in pro-inflammatory polarisation of macrophages and microglial cells during the resolution phase of the inflammatory response and, together with our cytokine profiling, this identifies CSPGs as important contributors to the pathological chronification of the inflammatory response after SCI that leads to irreversible tissue damage and resulting poor functional outcome.

Notably, we show that CSPGs directly convert distinct macrophage subsets to a non-resolution phenotype. To further understand the mechanisms behind CSPG immunomodulation we used BMDM and microglial cell cultures to evaluate the direct effects of CSPGs on innate immune cells. Recent studies have indicated that CSPGs can cause microglia to display proinflammatory properties and to adopt an M1-like phenotype[26]. Following stimulation of M2-polarised isolated microglia with

CSPGs, we found a significant increase in classical M1-associated proinflammatory mediators (IL1b, iNOS, TNFa, CCL3) in both adult and neonatal cultures, indicating a direct influence of CSPGs in blocking M2-like microglial conversion. Furthermore, in isolated macrophages (rat BMDM cultures), we discovered an even more potent and specific phenotype-dependent effect of CSPG stimulation which we hypothesise underscores our in vivo observations of inflammation resolution. We found that CSPG stimulation has minimal effect on already proinflammatory M1-like polarised macrophages. Interestingly, application of ChABC conditioned media reduced their basal expression levels of several inflammatory genes. This suggests that breakdown products as a result of CS-GAG digestion may positively modulate the inflammatory properties of M1-like macrophages. Indeed, CSPG disaccharides have previously been shown to be neuroprotective against excitotoxic damage[67] and digested oligosaccharide products have been shown to influence TLR4 pathway activation in M1-like macrophages[68]. In contrast to the limited effects of CSPG stimulation on proinflammatory M1 macrophages, and more meaningful in an inflammatory resolution context, we found that CSPG stimulation elicited potent effects on pro-resolving M2-like polarised macrophages, where they acted to phenotypically reverse the M2-like phenotype to M1-like, upregulating proinflammatory signature genes and reducing phagocytosis, essential for recovery of tissue homoeostasis[69]. These effects were attenuated by ChABC treatment (using ChABC enriched conditioned medium). Application of CSPG digestion product disaccharides did not induce inflammation. However, consistent with prior findings[70], chondroitin disaccharide Δdi-4S induced anti-inflammatory (CD206) marker expression. We cannot rule out that additional (indirect) effects of CSPGs may also modulate the inflammatory environment after SCI, for example by cytokine and chemokine retention[71,72]. However, these in vitro data provide evidence that CSPGs directly convert recruited monocyte/macrophages towards a M1-like phenotype and hinder the transition to M2-like at a tissue resolution level, resulting in chronification of the inflammatory response. CSPG digestion therefore represents a potent regulator of timely resolution of inflammation. Given the reciprocal activation of multiple cell types, such as astrocytes, by macrophages this further underscores a role of CSPGs in being potential central regulators of the neuroinflammatory response to injury.

We finally demonstrate that CSPGs convert the phenotype of macrophages by a TLR4-dependent mechanism. Having discovered that CSPGs block macrophage phenotypic conversion to pro-resolution, we examined whether TLR4 is involved in this CSPG immunomodulatory role. TLRs recognise a wide variety of pathogen-associated molecular patterns, initiate acute inflammation through the production of inflammatory cytokines[33] and play a pivotal role as an amplifier of the inflammatory response in "sterile" conditions. TLR4 activation by endogenous extracellular matrix (ECM) ligands has been explored in many inflammation and tissue injury paradigms. For example, biglycan is known to activate TLR4 in kidney injury[73] and sepsis, activation of TLR4 by fibronectin fragments is proinflammatory in myocardial infarction and stroke[74], and tenascin is a major TLR4 ligand in rheumatoid arthritis[75]. TLR4 activation by endogenous molecules has been less studied in the context of CNS pathology, but its enhanced expression in microglial cells and peripheral macrophages in neurodegeneration models[76] make it a candidate to contribute to disease progression in the absence of pathogens. Indeed, persistent exposure to danger signals can cause aberrant microglial activation and produce proinflammatory mediators, reactive oxygen and nitrogen species that propagate pathology in nervous system disorders[77]. After SCI, the role of TLR4 and its endogenous ligands is complex, with most reports showing

a detrimental inflammatory role of TLR4 activation[78] and improved recovery after TLR4 inhibition[79], although TLR4 inactivation has also been shown to increase astrogliosis and lesion pathology[80]. The role of CSPGs in TLR4 activation and its effect on the inflammatory response has not previously been established. Here, we provide data which links SCI upregulated CSPGs with TLR4 activation and detrimental consequences in inflammatory chronification. Using two approaches to inhibit TLR4 signalling (pharmacological inhibitors and a knockout model of TLR4), we show that TLR4 signalling is necessary for the CSPG-mediated phenotypic conversion of M0 and M2 macrophages towards a proinflammatory state. The significant upregulation of pro-inflammatory cytokines upon CSPG stimulation in M2 polarised BMDMs was consistently suppressed by TLR4 inhibition or deletion with the exception of CXCL10, which despite being linked with TLR4 activation by LPS, is mainly induced in response to IFN type-1 and type-2 receptor activation[81] and could be related with compensatory upregulation of other receptors (e.g. other TLRs) and activation of other pathways. Our results suggest that SCI upregulated CSPGs can act through TLR4 signalling in inactivated (M0) macrophages or when macrophages try to adopt a more repair phenotype (M2), causing the switch to an inflammatory phenotype and delaying the resolution phase of inflammation with devastating consequences.

Thus, we have identified a TLR4-dependent mechanism by which CSPGs exert effects on macrophages. CSPGs are known to signal through PTPσ, and previous work has shown this pathway can mediate inflammatory processes[26]. In agreement, we found that CSPGs do act on M1-like macrophages partially through PTPσ, since inhibiting PTPσ increased CD206 in M1-like macrophages. However, we demonstrate that TLR4 inhibition exerted far greater effects than PTPσ inhibition in the modulation of M2-like macrophages. As previously discussed, we found the effect of CSPGs on M1-like macrophages to be orders of magnitude lower than their role in converting M2-like macrophages to a more pro-inflammatory state. Here we find that TLR4 is critical to these effects, since TLR4 inhibition elicited a robust reduction in CSPG-induced pro-inflammatory gene expression in M2-like macrophages. In contrast, PTPσ signalling plays no clear role in M2 phenotypic conversion, since PTPσ inhibition did not reverse CSPG-induced pro-inflammatory gene expression in M2-like macrophages. Thus, CSPGs predominantly act via the TLR4 pathway to cause M2-like macrophages to switch to a proinflammatory phenotype. Furthermore, we show that CSPG–TLR4 pathway activation does not mediate other well-known functions of CSPGs, such as their growth inhibitory effects[3], since growth of cultured neurons in response to CSPG activation or degradation was uninfluenced by TLR4 signalling. This confirms immunomodulation as the critical role of CSPG–TLR4 interactions.

This work has implications for immunomodulatory therapies for SCI. Together our data indicate that CSPGs in the injured environment play a critical role at multiple stages of the immune response. First, they activate innate immune cells to a proinflammatory state— stimulating macrophages and microglia to express prototypical M1 pro-inflammatory markers. Second, they contribute to the infiltration of adaptive immune cells—propagating the expression of Th1 pro-inflammatory lymphocytes. Third, their continued presence delays inflammation resolution— while M1-activated immune cells should naturally convert to a pro-resolving M2 phenotype, CSPGs block this transition, keeping them in an activated pro-inflammatory state and perpetuating inflammation. These effects are mediated via TLR4 signalling. Further understanding of the mechanisms that prolong inflammation and hinder resolution will aid the development of pro-resolution strategies. CSPG-targeting strategies, such as ChABC[82]

and ISP[38] represent a particularly potent therapeutic approach for SCI, since they represent both a neuroplasticity and an immunomodulatory strategy, addressing two of the main goals for spinal cord repair[83]. Developing further anti-CSPG strategies could be an important avenue for treatment of other complex chronic inflammatory diseases.

In summary, we have identified a new role for CSPGs in resolution failure after SCI, where they prevent pro-resolution phenotypic conversion of immune cells via a TLR4-dependent mechanism. Insights into the dynamic interactions of CSPGs and immune cells may aid immunomodulatory therapies for SCI and other neurological disorders with a marked inflammatory component.

## Methods

**Animals**. One hundred and eighty-two adult female Lister Hooded (LH) rats (200–220 g; Charles River) were used for in vivo and in vitro studies. Rats were housed under a 12 h light/dark cycle with ad libitum access to food and water. All procedures were performed in accordance with the United Kingdom Animals (Surgical Procedures) Act 1986, approved by the Animal Welfare and Ethical Review Body (AWERB) of King's College London and conducted under Home Office Project License 70/8032 and PEE6F3C82. Methods and results are written in accordance with the ARRIVE guidelines for publishing in vivo research.

Twenty-four 10-week-old female mice C57/BL6 wild-type (WT; $n = 12$) and TLR4-Knock-out (TLR4$^{-/-}$; $n = 12$) (23–25 g; C57BL6 background kindly provided by Dr. S. Akira, Osaka, Japan) were used for in vitro experiments. Mice were housed under a 12-h light/dark cycle with ad libitum access to food and water, under controlled conditions of temperature (23 °C) and humidity (60%). All procedures were carried out in accordance with the guidelines approved by the European Communities Council Directive (86/609/ECC) and by Spanish Royal Decree 1201/2005 with the approval of the Ethical Committee of Animal Experimentation of the Príncipe Felipe Research Centre (Valencia, Spain).

**Lentiviral delivery of ChABC**. In order to express ChABC in the spinal cord, we used a lentiviral vector containing the cDNA coding for a mammalian-compatible engineered ChABC gene[28] (termed LV-ChABC), with ChABC expression driven by the mouse phosphoglycerate kinase (PGK) promoter. The production of these vectors is described in detail elsewhere[84] and second generation lentiviral vectors were generated as described previously[85,86]. The resulting vectors were integrating, self-inactivating and psuedotyped with VSV-G (vesicular stomatitis virus G). Viral particles were harvested by ultracentrifugation and titred by serial dilution of HEK293T cells followed by qPCR for the Woodchuck hepatitis virus post-transcriptional regulatory element (WPRE) as described previously[87]. A lentiviral vector generated from the same transfer vector containing the cDNA coding for GFP (termed LV-GFP) was used as a control, as described previously[24,87,88]. Both LV.PGK.ChABC and LV.PGK.GFP were titre-matched to $1.9 \times 10^{10}$ GC/ml.

**SCI model and treatment**. Adult female Lister Hooded rats ($n = 158$) were anaesthetised with isoflourane (O$_2$ 2 L/min) and skin was shaved then cleansed with sequential chlorohexidine and iodine swabs. Skin and overlying muscle were retracted and a thoracic laminectomy of T10 vertebral process was performed. Periosteum was removed. Rats then received a midline 150 kdyn spinal cord contusion injury at level T10 using an Infinite Horizon impactor and 2.5 mm diameter impact tip (Precision Systems Instrumentation). Immediately following injury, rats underwent midline intraspinal injection of viral vectors at two sites (0.5 μl, per site injected at 1 mm rostral and 1 mm caudal to the injury site), using a fine glass pulled-pipette and Microdrive pump (NanoLiter 2010 Injector/Micro 4 Controller, World Precision Instruments). The pipette was lowered 1.5 mm in the dorsoventral axis then retracted 0.5 mm and vectors were injected at a rate of 200 nl/min. The pipette was left in place for a further 2 min to ensure vector diffusion. Following intraspinal injection, overlying musculature and skin were sutured. Body temperature was maintained at 37 °C using a self-regulating heating mat throughout and animals recovered for at least 1 h in an incubator (water base thermostat 32 °C, Thermocare) and were then returned to home cages. Analgesia (Caprieve, 5 mg/kg) was administered peri-operatively, and animals received daily subcutaneous injection of 5 ml of saline, 5 mg/kg of Baytril and 5 mg/kg of Caprieve for 3 days following surgery. Bladder expression was performed twice a day until reflexive bladder emptying was restored (typically by 7 dpi). In the acute postoperative period following surgery, extensive welfare checks were carried out, including provision of accessible chow. hydration gel and soft bedding in all cages, before returning to standard husbandry conditions.

**Bone marrow-derived macrophage culture (BMDM)**. BMDMs were obtained from the tibia and femurs of LH rats and naïve C57BL6 or TLR4$^{-/-}$ mice. Animals were euthanized by CO$_2$ and bone marrow was flushed from the bones with ice-cold sterile DMEM medium (Gibco$^{TM}$) using a 23 G needle. The suspension was

filtered through 70 μm cell strainers (Falcon) and red blood cells were lysed. Immature monocytes were cultured at a concentration of 200,000–250,000 cells/cm$^2$ in DMEM+ Glutamax (Gibco$^{TM}$) supplemented with 10% foetal bovine serum (Gibco$^{TM}$), 50 units/ml of penicillin/streptomycin (Pen-Strep; Thermo Fisher, 15140122) and 25 ng/ml (500 U/ml) macrophage colony-stimulating factor (M-CSF, Peprotech) at 37 °C in a water saturated atmosphere with 5% CO$_2$. Cells were fed with M-CSF containing media every 3 days and harvested after 6 days in culture by incubating with Enzyme-Free cell dissociation buffer (Millipore) for 15 min at 37 °C. Cells were gently centrifuged and resuspended in complete medium (DMEM with high glucose and 10% foetal bovine serum and Pen-Strep) and were seeded alone or in poly-D-lysine-coated plates at a density of 1–1.2 × 10$^5$ cells/cm$^2$. After 24 h, cells were polarised using recombinant IL-4 (20 ng/mL, Peprotech)-supplemented complete medium for anti-inflammatory polarisation (M2), or lipopolysaccharide (LPS 100 ng/mL, Enzo Life Sciences) and IFNγ (20 ng/mL, Peprotech) supplemented complete medium for proinflammatory polarisation (M1). Non-polarised controls (M0) were incubated for the same amount of time with complete medium. Cells were cultured for 24 h. Flow cytometry analysis were performed to establish the purity of rat BMDM, as described below. Polarisation of rat BMDM and mouse BMDM were evaluated by qPCR and flow cytometry or qPCR analysis, respectively, as described below.

**Neonatal microglial cell culture**. Primary microglia were obtained from mixed glia cultured from the forebrain of 2-day-old LH rat pups. Postnatal pups were anesthetised on ice and rapidly decapitated. Brains were isolated in Dulbecco's PBS (DPBS, Gibco$^{TM}$) on ice and stripped of olfactory bulbs, cerebellum and midbrain and meninges. Forebrains were minced using a scalpel blade and the tissue pieces were centrifuged at 300 g for 5 min, the supernatant removed and 1 ml of 0.25% Trypsin EDTA (Gibco$^{TM}$) with DNAseI (200 μg ml$^{-1}$; Roche Diagnostics Gmb) per pup was used to resuspend the pellet, which was incubated (37 °C) for 20 min, triturated and passed through a 70 μm cell strainer to obtain a single-cell solution. 10 ml of pre-warmed DMEM/F12 complete medium (DMEM/F12,Gibco$^{TM}$) supplemented with 10% foetal bovine serum (FBS, Gibco$^{TM}$) and 50 units/ml of Pen-Strep was added to the pellet. The cell suspension was centrifuged (400 × g, 5 min at 4 °C), resuspended in complete medium, filtered through a 70 μm cell strainer and seeded at (300,000 cells/cm$^2$) into 75 cm$^2$ T-flask (Corning) coated with poly-D-lysine (PDL) (0.1 mg/ml. Sigma-Aldrich). Cells were incubated at 37 °C in a water saturated atmosphere with 5% CO$_2$, with 50% of the total media refreshed at day 3 and 7. Upon cell confluence (10–12 days in culture), flasks were shaken for 5 h at 300 rpm at 37 °C to harvest microglial cells and seeded at 100,000 cells/cm$^2$ concentration. Microglia culture purity was evaluated by qPCR, described below.

**Adult microglial cell culture**. Adult microglia were isolated based on a modified version of an established protocol[89] LH rats were deeply anaesthetised with sodium pentobarbital (Euthatal$^®$, 80 mg/kg, administered intraperitoneally) and transcardially perfused with ice-cold DPBS containing 2 mM EDTA. Brains were dissected, homogenised with Dounce homogeniser in 5 ml ice-cold DPBS, filtered through a 70 μm cell strainer to remove any debris and centrifuged (300 × g, 7 min, 4 °C). Pellets were resuspended in 7 ml of 30% Percoll$^®$ (Sigma, P1644) and centrifuged (800×g, 30 min, room temperature), to remove myelin. Microglia were isolated from the pellet with CD11b conjugated magnetic microbeads (Miltenyi Biotec) according to manufacturer's protocol. Magnetic-activated cell sorted CD11b+ cells were flushed from LS columns (Miltenyi Biotech) and collected in complete medium DMEM/F12 supplemented with 10% FBS and Pen-Strep. Cells were seeded at 100,000 cells/cm$^2$ concentration in 24-well plates coated with poly-D-lysine (0.1 mg/ml. Sigma-Aldrich).

**In vitro stimulation of microglia and macrophages**. M0 macrophages were incubated with 1, 2.5 and 5 μg/ml soluble CSPGs (Merck-Millipore, cc117) for 4 h to determine the optimal CSPG dose, evaluated by qPCR. M0, M1, and M2 macrophages were incubated with CSPG on a PDL+ CSPG matrix or in a solubilized form for 4 h to determine the optimal method for CSPG stimulation. For CSPG immunocytochemistry, coverslips were fixed with 2% PFA for 8 min, washed 3 times with PBS and incubated overnight at 4 °C with mouse anti-CSPG (Supplementary Table 4—AbD Serotec; 1:250) in PBS with 5% donkey serum. Following PBS washes, coverslips were incubated for 1 h in secondary antibody solution (AlexaFluor-488 anti-mouse 1:1000 in PBS) and mounted with Vectashield containing Dapi (Vector Laboratories). Images were acquired using Zeiss Axioplan 2 fluorescence microscope. To evaluate immunomodulatory effects of CSPGs, M0, M1, and M2 polarised macrophages and neonatal and adult microglial cells were cultured with complete medium supplemented with or without CPSGs (5 μg/ml) for 4 or 16 h at 37 °C in a water saturated atmosphere with 5% CO$_2$ and the effect of CSPG on BMDM cultured cells was evaluated by flow cytometry, immunoblotting and qPCR analysis, described below. Immunomodulatory role of CSPG digestion products assay, performed following the CSPG activation protocol. M1 and M2 polarised macrophages were cultured with complete medium supplemented with or without Chondroitin disaccharide Δdi-0S and Δdi-4S (DextraUK—C3201 and C3202) (5 μg/ml) for 4 at 37 °C in a water saturated atmosphere with 5% CO$_2$ and the effect of CSPG on BMDM cultured cells was

evaluated by qPCR analysis. Finally, to evaluate the role of the receptors TLR4 and PTPσ and the MAPK p38 pathway in CSPG inflammatory activation M1 and M2 polarised macrophages were cultured with TAK242 (2.5, 10 and 100 μM—Cayman Chemical 13871), ISP (2.5 and 10 μM—PTPσ Inhibitor, ISP—Calbiochem—Sigma Aldrich 5343390001) or SB202190 (25 μM—Sigma Aldrich S7067) respectively, 1 h prior to CSPG incubation.

**Phagocytosis assay**. For the phagocytosis assay, green fluorescent latex beads (Sigma, L4655) were diluted in cell culture media and added to macrophage cultures at a final concentration of 0.02% (v/v). After 90 min, media was removed, and cells were washed three times with PBS then fixed in 4% PFA for 10 min. For cell body staining, the coverslips were incubated in Phalladin-568 for 30 min then mounted in Fluoromount-G with DAPI. Five randomly distributed frames per coverslip were acquired using an LSM710 confocal microscope and the average number of cells containing fluorescent beads was determined.

**Generation of ChABC-conditioned media and CSPG digestion**. Differentiated rat BMDM cells were plated on PDL precoated glass coverslips at 100,000–120,000 cells/cm$^2$ density in complete media. After 16 h, media was replaced with viral infection media (complete media supplemented with either LV-GFP or LV-ChABC, used in the in vivo experiments, at different concentrations (0, 2*10$^7$, 4*10$^7$ and 1*10$^8$ GC/ml) and incubated overnight. Following washes with DMEM, cells were incubated with complete medium for 48 h prior to collection of ChABC or GFP conditioned media. BMDM transfection efficiency was determined by both qPCR (described below) and by quantifying the percentage of GFP-positive cells transduced by LV-GFP. Cells were fixed and the GFP signal amplified using immunocytochemistry (Supplementary Table 4—chicken anti-GFP primary antibody [Abcam,1:1000], AlexaFluor-488 anti-Chicken [1:400] secondary antibody). Images were acquired using Nikon A1R Si Confocal Imaging system on an Eclipse Ti-E inverted microscope. Optimised conditioned media was used for further in vitro CSPG digestion experiments. For determining ChABC effects on polarised BMDMs, M0, M1 and M2 polarised macrophages were cultured for 4 h with complete medium derived from cells expressing GFP or ChABC enriched medium, both alone or supplemented with CSPGs (5 μg/ml, soluble). After incubation, medium was removed and after three washes cells were collected. The effect of CSPG digestion was assessed by qPCR.

**Endocytic TLR4 quantification**. M1 or M2 mouse derived macrophages were fixed with PFA 4%, permeabilized or not with 0.1% Triton X-100 and subsequently blocked with 5% normal goat serum in PBS. The primary antibody against TLR4 (Supplementary Table 4) incubated overnight at 4 °C. After washing, AlexaFluor-488 conjugated antibody (1:400, Invitrogen) against mouse IgG was incubated for 2 h at R.T. All cells were counterstained by incubation with 4,6-diamidino-2-phenylindole dihydrochloride (DAPI; Invitrogen). Apotome Zeiss microscope was used for image acquisition at ×63 magnification. Image J was used for fluorescence signal quantification. After selecting a colour threshold to include the whole and individual cells, excluding particles smaller than 100 mm, and obtaining the averaged fluorescence signal per cell.

**Axonal length quantification**. Primary cultures of cortical neurons from wild type or TLR4$^{-/-}$ mice foetuses at 14 days of gestation (E14.5) were generated as previously described[90]. Briefly, brain cortex was mechanically triturated and once homogeneous, filtered through a nylon net with 90 μm pores and centrifuged at 500 rcf for 5 min. Cell pellets were first seeded into DMEM supplemented with 10% foetal bovine serum (FBS) and 1% P/S into poly-L-lysine (PLL)-coated plates (plating medium). After 2 h of incubation at 37 °C, 5% CO$_2$, the plating medium was replaced by Neurobasal medium supplemented with 1X B27 supplement (Gibco), 50 mM Glutamax (Gibco) and 1X P/S. Three days after plating, CSPG (5 μg/ml) or its vehicle alone or with ChABC (0.1 IU) were added to the cell medium and incubated for 24 h. Cultures were fixed with 4% paraformaldehyde (PFA) in PBS for 10 min, then permeabilized and blocked with 5% normal goat serum (NGS; Thermo Fisher) and 0.2% Triton X-100 (Sigma). Cells were incubated overnight at 4 °C with primary antibodies against β-III-tubulin (described in Supplementary Table 4). AlexaFluor-488-conjugated antibody (1:400, Invitrogen) against mouse IgG was used. DAPI was used to stain cell nuclei. Fluorescent images were acquired using a fluorescent microscope (Apotome; Zeiss, Oberkochen, Germany). Consistent exposures were applied for all images, and ImageJ software was used for image analysis. The NeuronJ plugin was used to quantify neurite and axon lengths as previously described[91]. Between 120 and 175 neurons were analysed per condition, in triplicates.

**Flow cytometry sample preparation**. To study the dynamics of immune cells after SCI, spinal cords from sham (laminectomy only) and injured LV-GFP or LV-ChABC treated rats were harvested at day 1, 3, 7, 14 and 28 days after lesion. Animals were deeply anaesthetised with sodium pentobarbital (Euthatal$^®$, 80 mg/kg, administered intraperitoneally) and transcardially perfused with ice-cold 1X phosphate buffered saline (PBS) + 2% EDTA. Immediately after perfusion, 8 mm of the injured spinal cord centred around the lesion epicentre was dissected and placed into ice-cold PBS. Tissue was mechanically dissociated and then passed

through a 70 μm cell strainer (BD Falcon, Germany), and centrifuged at $300 \times g$ at 4 °C. The pellet was incubated with Myelin Removal Beads II (Miltenyi Biotec, Germany) and passed through LS columns (Miltenyi Biotech) to elute cells. For primary cell cultures, cells were detached by incubating with Enzyme Free cell dissociation buffer (Milipore). Remaining adherent cells were gently scraped and passed through a 70 μm cell strainer (BD Falcon, Germany). After centrifugation, cells were resuspended in DPBS (Gibco).

**FACS staining/gating and analysis**. Both isolated CNS cells and primary cell cultures were washed with cold PBS then incubated with a live/dead stain (eBioscience). After cell counts, samples were incubated with anti-rat CD16 and CD32 (1:50, BD Bioscience) for 15 min on ice to block the Fc receptors and stained with specific extracellular antibodies for 30 min (antibodies are detailed in Supplementary Table 1—Panel 1). For TLR4 expression studies, cells were stained with CD45, CD11b, and TLR4 (detailed in Supplementary Table 1—Panel TLR4 detection). For intracellular staining used in phenotype analysis experiments, cells were then washed, fixed using 2% PFA and permeabilized with cell permeabilization buffer (Invitrogen) containing intracellular antibodies (antibodies are detailed in Supplementary Table 1—Panel 2). Single-stained Compensation Beads (BD) were used according to manufacturer's instructions to prepare compensation controls by incubating with fluorescently conjugated antibodies used in the experiments. Fluorescence minus one (FMO) experiment and isotype-matched control samples were run prior to this study to establish the positiveness of the samples and to aid the optimisation of the compensation matrix. Based on this, the compensation matrix was adjusted where necessary due to over- or under-compensation by the automated algorithm.

Cells were acquired on LSRFortessa III flow cytometer (BD) and the data was analysed with FlowJo (V10, Treestar) software. Single live cells were gated on the basis of dead cell exclusion (L/D), side (SSC-A) and forward scatter (FSC-A) gating, and doublet exclusion using side scatter width (SSC-W) against SSC-A. To perform the analysis, cells were first gated for CD45 to ensure that only infiltrating leucocytes and resident microglia were selected. Then, a combination of markers were used to identify the following cell populations: microglia (CD45+$^{medium}$, CD11b+$^{medium/high}$, CD3 and CD19$^{low}$ and SSC$^{low/medium}$), macrophages (CD45+ $^{high}$, CD11b+$^{high}$, CD3 and CD19$^{low}$, and SSC$^{low/medium}$), Neutrophils (CD45+$^{high}$, CD11b+$^{medium}$, CD43+$^{high}$, HIS-48+, RP1+, CD3 and CD19$^{low}$ and SSC$^{high}$), CD4+(CD45+$^{high}$, CD11b− or low, CD3−, CD4+, CD8−), CD8+ T cells (CD45+$^{high}$, CD11b− or low, CD3+, CD4−, CD8+) and B cells (CD45+$^{high}$, CD11b− or low, CD11c−, CD3−, CD45RA+). To study the phenotype of microglia and macrophages, in addition to prior described antibodies, these cells were further differentiated based on CD86, CD68, MHC II, iNOS, CD43, HIS48, CD163, CD206, and Arg1 expression. The FACSymphony A5 flow cytometer, equipped with UV (355 nm), violet (405 nm), blue (488 nm), yellow/green (561 nm), and red laser (637 nm) was utilised for these experiments. Data was analysed with FlowJo (V10, Treestar) software and population clustering was performed by t-SNE FlowJo Plugin.

**t-SNE analysis**. The complex maps of immune cells were plotted by t-distributed stochastic neighbour embedding (t-SNE)[92], which reduced dimensionality of multi-colour flow cytometry data into a 2-dimensional data space (tSNE-1 vs. tSNE-2). Concatenating graphs are generated from all samples in each group. Manually-gated viable CD45+ leucocytes were overlaid into the tSNE plots using FlowJo plugin (version v10-LLC) and clustered by relative marker expression into nodes using the following parameters: perplexity = 50, theta = 0.5 and 500 iterations.

**Cell sorting**. Extracellular staining was performed in 7 dpi spinal cord isolated cell samples. After Fc-receptor blockade and live/dead staining, specific antibodies were used to identify neutrophils, microglia, CD43−/CD43+ monocyte/macrophages and T CD4 populations (antibodies detailed in Supplementary Table 1). Cells were isolated using an Aria III (BD Bioscience) and sorted cells were collected into RLT lysis buffer (Qiagen) with 1% β-mercaptoethanol and frozen at −80 °C.

**Cytokine and chemokine expression analysis**. Animals were deeply anaesthetised with sodium pentobarbital (Euthatal®, 80 mg/kg, administered intraperitoneally) and transcardially perfused with sterile PBS supplemented with EDTA (12.5 mM). An 8 mm section of contused injury epicentre spinal cord (or equivalent intact control) was collected at 6, 12 h and at 1, 3 and 7 days after contusion and snap-frozen in liquid nitrogen. Spinal cords were bisected while frozen, in order to conduct both protein and RNA analysis (described below). For protein analysis, tissue was homogenised with lysis buffer on ice with a tissue homogeniser (Cole-Parmer, LabGEN 7 Series Homogeniser). Samples were centrifuged ($20,000 \times g$, 4 °C, 20 min), the supernatant recovered, and protein quantified using the Pierce BCA Protein Assay Kit (Thermo Fisher Scientific). Samples were concentrated to 4 mg/ml using MicroCon centrifugation filters (Millipore) to ensure equal amounts of protein. Cytokine protein levels were then analysed using the MILLIPLEX MAP Rat 27 Cytokine & Chemokine magnetic bead panel (RECYMAG65K27PMX—Millipore) on a Luminex (Millipore) as per manufacturer's protocol.

**Immunohistochemistry**. Rats were anesthetized with an overdose of pentobarbital and perfused with PBS and 4% paraformaldehyde (PFA). Spinal cords were dissected and post-fixed in 4% PFA overnight at 4 °C, rinsed with PBS and dehydrated in 20% sucrose. 20 μm-thick sections were mounted on SuperFrost slides after cryosectioning. For iNOS staining, heat mediated antigen retrieval was performed prior to blocking by incubating the slides in citrate buffer pH 6.0 at 90 °C for 30 min. After blocking, sections were incubated with primary antibodies overnight at 4 °C and followed by appropriate secondary antibodies. Primary antibodies used are detailed in Supplementary Table 4. The sections were incubated with appropriate fluorescently conjugated secondary antibodies: goat anti-mouse IgM AF-568 (#A21043, Thermo, 1: 500), donkey anti-rabbit IgG AF-568 (#A10042, Thermo, 1:500) and coversliped using Fluoromount-G with DAPI mounting medium. For each staining, four individual animals per group were examined and images were captured with a ZEISS Imager Z1 fluorescence microscope equipped with a AxioCam MRm camera. Tile scans were stitched in post-acquisition processing using AxioVision software. Staining was quantified using ZEISS ZEN software by measuring the mean fluorescence over sections excluding roots and meninges.

**Immunoblotting**. Spinal cord samples used for the Luminex assay were also used for Western blotting. Protein extracts were denatured and reduced in 2× sample buffer (500 mM Tris, pH 6.8, 40% glycerol, 0.2% SDS, 2% β-mercaptoethanol, and 0.02% bromophenol blue), and boiled (98 °C, 10 min). For BMDM immunoblotting, proteins were extracted with radioimmunoprecipitation assay (RIPA) buffer (Sigma-Aldrich) on ice for 30 min. Samples were centrifuged ($20,000 \times g$, 4 °C, 20 min), the supernatant recovered, and protein quantified using BCA as above.

Twenty micrograms of protein per sample were loaded and separated on 15-well, Bis–Tris 4–12% polyacrylamide gels (NuPAGE, Invitrogen). Proteins were then transferred on nitrocellulose membranes (ThermoFisher Scientific) for immunoblotting. Membranes were stained with Ponceau red stain to visualise protein transfer and to confirm equal protein loading. Following blocking in 5% fat-free milk powder in PBS, membranes were incubated overnight at 4 °C with anti-rat or anti-mouse antibodies against phosphorylated protein form described in Supplementary Table 3. Following three washes in PBS Tween 20, membranes were incubated with the appropriate horseradish peroxidase-conjugated secondary antibodies (Dako Cytomation) in 5% fat-free milk powder at 1:2000 dilution (1 h, room temperature). Enhanced chemiluminescence (ECL) reagents (GE Healthcare) and a Kodak processor and Alliance Q9 Advanced (Uvitec Cambridge) were used for detection and blot development. Membranes were restored using western blot stripping buffer (Thermo Scientific) and re-blocked in 5% BSA in PBS. The membranes were incubated (4 °C, overnight) with anti-rat antibodies against total form of proteins described in Supplementary Table 3. Bands were detected using chemiluminescence as described above. Densitometry of developed blots was performed using ImageJ (https://imagej.nih.gov/ij/) and values were normalised to β-actin or GAPDH.

**pP38 expression by immunofluorescence analysis**. M1 or M2 mouse derived macrophages were incubated with CSPG (5 μg/ml) or its vehicle for 24 h, and then were fixed with PFA 4%, permeabilized with 0.1% Triton X-100 and subsequently blocked with 5% normal goat serum in PBS. The primary antibody against or p38 (described in Supplementary Table 4) were incubated overnight at 4 °C. After washing, AlexaFluor-555 conjugated antibody (1:400, Invitrogen) against rabbit IgG was incubated for 2 h at RT. All cells were counterstained by incubation with 4,6-diamidino-2-phenylindole dihydrochloride (DAPI; Invitrogen). Apotome Zeiss microscope was used for image acquisition at ×63 magnification. Image J was used for fluorescence signal quantification. After selecting a colour threshold to include the whole and individual cells, excluding particles smaller than 100 mm, and obtaining the averaged fluorescence signal per cell.

**RNA extraction and reverse transcription**. Total RNA from frozen spinal cord and sorted immune cells were extracted using the RNeasy Mini Kit including DNaseI treatment (Qiagen) according to the manufacturer's recommendations. For primary cultured cells, cells were homogenised in TRIzol® reagent (Thermo Fisher Scientific). An aqueous (RNA-containing) phase was generated using 1:5 bromo-chloro-propane, mixed 1:2 with 70% isopropanol and centrifuged at $12,000 \times g$ to precipitate RNA. Samples were treated with DNaseI (Qiagen). RNA concentration and integrity were determined by a NanoDrop ND-1000 spectrophotometer (Thermo Scientific). Total RNA with an A260/A280 ratio ranging from 1.7 to 2.2 were converted to cDNA using the high capacity RNA-to-cDNA™ kit (Applied Biosystems).

**qPCR**. Quantitative PCR (qPCR) primers (Sigma) were designed using Primer-BLAST (NCBI; sequences are detailed in Supplementary Table 5). A total of 10 ng of cDNA was used for quantitative PCR in a total volume of 10 μl with LightCycler 480 SYBR Green I Master Mix (Roche, Switzerland) and specific primers (detailed in Supplementary Table 5), on a LightCycler 480 (Roche, Switzerland). The amplification conditions were determined by the primers to present amplification efficiency close to 100% and a single peak in melt-curve analyses. In sorted cell samples, genes of interest were pre-amplified by TaqMan™ PreAmp Master Mix kit (Applied Biosystems) following manufacturer´s instructions. Each Real-time PCR

reaction was performed in triplicates. Glyceraldehyde 3-phospate dehydrogenase (GAPDH) and Actb (β-actin) were used as housekeeping gene. The log fold change in mRNA expression was calculated from ΔΔCt values relative to control samples.

**Multidimensional analysis for gene expression.** Dual multiple factor analysis (dMFA)[93] was conducted to study the longitudinal evolution of the multi-dimensional gene expression profile. Different groups of subjects (animals at different timepoints) were analysed on the same variables (mRNA levels of selected genes). Global patterns (consensus components or dimensions across time) were extracted and the effects of time in the gene expression patterns (time partial components or dimensions) were studied. Loadings (relationship of genes with the components) are interpreted as the correlation of a variable with a component (ranging from −1 to 1) where the value of each animal for each component (component scores) was extracted and dMFA performed using the R package FactoMineR[94], with scaling gene values to unit variance on samples at 6h, 12 h, 1, 3, 7 and 14 days post-injury. Three Cattel's criteria global components were determined as representative by Scree test[95]. |loading| ≥ 0.3 are marked as the cutoff for the relative importance of each component on each gene. To study the impact of time and experimental group on the three extracted components, a two-full factorial ANOVA was performed taking time, group and their interaction as "factors" for each component and "group by time pairwise comparison" was performed using Tukey correction for multiple comparisons. The Euclidean distance to naive centroid (average) for the three dimensions was calculated for each animal. A two-full factorial ANOVA was also conducted in these distances as described above. At 7 dpi (a key resolution time point), a second multidimensional analysis was conducted using PCA. Cattell's criteria selected three components as before. A two-full factorial ANOVA was conducted (taking group, component and the interaction between them as factors), with alpha significance level of 0.05. Loading and score plots were generated using the ggplot2 R package[96].

**Statistical analysis.** Numerical values are reported as mean ± standard error of the mean (SEM). Datasets were tested for normality using Kolmogorov–Smirnov tests. Student's *t*-test was used for single comparisons between two groups and one-way or two-way ANOVA followed by Bonferroni's multiple-comparison tests for more than two groups when the data were normally distributed. Mann–Whitney tests were performed for two-sample comparisons or Kruskal–Wallis tests with Dunn's post hoc comparisons for more than two samples if the data were not normally distributed. Normal distribution was assumed if sample size was sub-threshold for normality testing. Statistical analyses were performed with GraphPad Prism v8 and v9 software and differences were considered significant * at $p < 0.05$, ** at $p < 0.01$ and *** at $p < 0.001$. All statistics and post hoc tests are stated in the figure legends and correction for multiple comparisons performed where appropriate. Multivariate analyses (dMFA and PCA) were performed in R (package FactoMineR).

**Blinding and randomisation.** During surgical procedures blinding was ensured by the experimenter injecting the vectors being unaware of animal identification number. Animals across all groups were randomised into cages. Experimenters were also blinded to treatment groups for histological analyses. For in vitro experiments, activation, data collection and all statistical analysis was completed with the investigator blind to the experimental coding. In some cases, such as TLR4 WT vs. KO experiments, blinding was not possible during cell isolation. However, as in the rest of in vitro experiments, activation, data collection and all statistical analysis was completed with the investigator blind to the experimental coding.

**Figure design.** Figures were designed using GIMP 2.10.22 and Biorender (with full licence to publish).

**Reporting summary.** Further information on research design is available in the Nature Research Reporting Summary linked to this article.

## Data availability
Complete source data are provided with this paper. Data that support the findings will be made available upon reasonable request to the corresponding author.

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

## Acknowledgements

The work was supported by grants from the following organisations: the U.K. Medical Research Council (SNCF G1002055; ERA-NET NEURON MR/R005532/1) and the Rosetrees Trust (CF1\100006) to E.J.B.; the Fondo Europeo de Desarrollo Regional (FEDER)/Ministerio de Ciencia e Innovación—Agencia Estatal de Investigación "RTI2018-095872-B-C21/ERDF" (included in the FEDER programme for the Comunidad Valenciana 2014-2020) to V.M.M.; and Wings for Life Spinal Cord Research Foundation (WFL-UK-01/20 Project 214 to E.J.B.; WFL-NL-25/20 Project 238 to J.V.). We thank Dr. S. Akira (Osaka) and Dr C. Guerri for providing TLR4$^{-/-}$ mice.

## Author contributions

E.J.B. and I.F.Q. conceived the study. I.F.Q. carried out the majority of the experiments. I.F.Q., L.M., and E.R.B. performed the rodent surgeries. J.V. and F.D.W. generated viral vectors for the study. I.F.Q. performed the FACS/Flow experiments, T-SNE analysis, qPCR, Western blot and Luminex studies. I.F.Q. and A.T.E. analysed gene expression data. I.F.Q. and M.S.P. performed the culture studies. M.S.P. and V.M.M. performed the TLR4 KO studies. S.B. performed the histological procedures. I.F.Q., M.S.P., V.M.M., A.T.E. and S.B. analysed the data. I.F.Q., M.S.P. and S.B. generated the figures. E.J.B., I.F.Q. and E.R.B. wrote the paper. E.J.B., V.M.M. and J.V. secured funding.

## Competing interests

The authors declare no competing interests.
