## [Peer Review File · Nature Communications]

Reviewers' Comments:

Reviewer #1:

Remarks to the Author:

This study by Quijorna et al. shows that CSPGs, in addition to their well-characterized role in preventing axonal regeneration, are also involved in the recruitment of immune cells and the promotion of a proinflammatory phenotype at sites of SCI. Notably, the authors found that digestion of CSPG glycosaminoglycan side chains by means of lentivirus-mediated expression of the bacterial enzyme chondroitinase ABC (ChABC) enhanced immune cell clearance and reduced pro-inflammatory gene and protein expression profiles in the injured rat spinal cord during the subacute phase of injury. Importantly, they demonstrated that the proinflammatory effects of CSPGs are mediated through TLR4. Overall, this paper is well-written and the data are convincing using the most rigorous and unbiased methods possible. The study is relevant and important as it may contribute to explain why tissue damage is reduced and functional recovery improved in SCI rats following ChABC treatment (Bartus et al., 2014). I only have a few minor issues with this paper, which are enumerated below and should be taken into consideration in a revised manuscript.

- 1) The overall message is a bit confusing when it comes to which macrophage population is affected by CSPGs in vivo (Fig. 3) versus in vitro (Fig. 5). Please also see Comment #3, below.
- 2) According to Fig. 2d and Extended Figs. 3-4, although this is true that the expression of inflammatory-related genes returned to naive levels in lentivirus-ChABC treated rats compared to lentivirus-GFP treated animals at day 7 post-SCI, it is also true that signs of inflammation were higher with ChABC gene therapy at day 3. Please comment on the possibility that the temporal dynamics of inflammation and its resolution may just have been slightly shifted in time in response to ChABC treatment.
- 3) According to the authors' hypothesis, CSPGs that are produced during the subacute phase of SCI act on TLR4-expressing macrophages and microglia to maintain (or boost) their proinflammatory status, thus preventing inflammation resolution and maintaining a state of chronic inflammation at sites of SCI. However, the data presented in Fig. 5 suggest that CSPGs mainly act on M2 macrophages with very little effects on M1 macrophages, which seems counterintuitive. Could this have to do with the way that macrophages were polarized towards an M1 state in vitro? Indeed, M0 macrophages were turned into M1 macrophages upon treatment with LPS, a potent TLR4 agonist that leads to the activation of the LPS/CD14/TLR4 complex and its subsequent endocytosis by endosomes (this has been shown by many, including: Tan et al., *Immunity*, 2015; Rajaiah et al., *PNAS*, 2015). Could the lack of effect of CSPGs on M1 macrophages be due to their lower expression of membrane-bound TLR4 resulting from LPS-mediated TLR4 endocytosis? At the very least, TLR4 expression should be assessed in the various subsets of macrophages under investigation in culture.
- 4) Please indicate whether the results suggesting neutrophil clearance following ChABC treatment through a comparison of resolution rates are significant or not (with supporting statistics).
- 5) Can the proinflammatory effects of CSPGs on macrophages/microglia be recapitulated by biglycan alone?
- 6) A key element that is missing are experiments that would prove that CSPG receptors other than TLR4, for example LAR and PTP σ , do not play a redundant role in this proinflammatory function of CSPGs, as suggested by others (see Dyck et al., 2018). In M1 macrophages that are only partially activated, could CSPGs elicit their proinflammatory effects via LAR or PTP σ ? Was the phosphorylation of p38 maintained in these conditions (TLR4 deficiency) as a result of CSPG stimulation?

Reviewer #2:

Remarks to the Author:

In this manuscript, Francos-Quirorna et al describes their interesting results showing that enzymatic digestion of CSPG glycosaminoglycans facilitates inflammation resolution in the spinal cord lesions. This is supported by the results from multiple non-biased and elegant analyses of immune cells in spinal cord lesions. Furthermore, they implicate TLR4 as a critical player in mediating the effects of CSPGs. Together, these results reveal a novel mechanism that regulate inflammatory response in spinal cord lesion and might have translational potential. I have only a few minor suggestions for the authors to consider during revision.

While in vitro studies suggested a direct role of CSPG in regulating immune cell responses, it cannot formally rule out the possibility that CSPGs might have an indirect role. In fact, early studies showed that CSPGs could bind different soluble factors in the organization of matrix-bound cues in the brain (for example, Emerling and Lander, 17, 1089, 1996). Thus, an alternative possibility is that CSPGs sequester (pro- or anti-) inflammatory factors generated in the lesion and ChABC treatment releases these factors, contributing to the effects reported in this manuscript.

In this regard, what is the fate of glycosaminoglycans after enzymatic digestion? How long do these cleaved products remain in the lesion? Do they still have the ability of activating TLR4?

Reviewer #3:

Remarks to the Author:

Francos-Quirorna et. al. conducted detailed characterizations of the effects of CSPG digestion on the immune contexture in a rat spinal cord injury model, and attributed these changes to TLR4-mediated pro-inflammatory phenotypes of microglia/macrophages. The authors did a careful temporal comparison of the abundance of different immune cells at the injury sites using both manual flow cytometry and a t-SNE flow cytometry analysis, which showed reduced number of immune cells specifically at 7 day post-injury. They then probed inflammatory gene expression at whole tissue level and in individual cell populations. Finally, they conducted in vitro studies to show a direct effect of CSPG on upregulation of pro-inflammatory genes in M2-primed macrophages, an effect that is dependent on TLR4. Although the studies were comprehensive, the data were largely descriptive with no significant conceptual advancement for the field, and there are a number of shortcomings.

Major critiques:

1. Even though authors presented some new data linking CSPG and pro-inflammatory immune cell phenotype, overall the study falls short of conceptual advancement, as a number of recent studies, including PI's own work, have already shown that CSPG modulates immune response and neuroinflammation in many CNS disorders, including spinal cord injury. Along the same line, the current study has been conducted within the conceptual framework of pro- vs. anti-inflammatory immune responses, while in reality, there is considerable heterogeneity one single cell level in macrophages or microglia after CNS injury, with many cells express both anti- and pro-inflammatory genes. Hence, the role of CSPG in promoting pro-inflammatory immune responses might be an over-simplification. In fact, many inflammatory genes examined here did not consistently show changes that fit with the simplified model (see comment below).
2. It appears that authors only highlighted the results that "fit with their model", while ignoring results that do not fit or even appear opposite. For instance, data in Fig. 2d showed opposite results at 3 dpi and 7 dpi: at 7 dpi, there was a decrease in pro-inflammatory cytokine gene expression in LV-ChABC group than in LV-GFP group, while at 3 dpi, the opposite is true. The same is also true for many pro- or anti-inflammatory genes.
3. Another major shortcoming of the study is the lack of spatial information of the immune contexture at the injury site with or without CSPG digestion. The current study relies heavily on flow cytometry, which only revealed the abundance of different immune cells without information regarding their localization, spatial pattern, interaction with astrocytes or axons, and local matrix composition.

4. There is also a missed opportunity to compare axon regeneration and functional recovery in TLR4^{-/-} mutant mice vs. mice treated with LV-ChABC. If TLR4 functions mainly as a receptor for CSPG in promoting pro-inflammatory phenotypes of microglia/macrophages, TLR4 KO would phenocopy LV-ChABC. More interesting perhaps is to conduct SCI study combining TLR4 KO and LV-ChABC, which would will provide crucial in vivo genetic evidence linking CSPG and TLR4. It will be interesting to determine if combination would result in no further gain in reducing inflammatory milieu, although CSPG may function through other receptors or on other cells such as astrocytes.

Additional comments:

1. Fig. 1: After SCI, macrophages continue to arrive at the injury site, while early-arrived macrophages are being cleared. Hence the reduced number of macrophages at 7 dpi may also reflect a reduction of new recruitment of monocytes/macrophages, and not necessarily clearance. Without lineage tracing experiments, the data do not support the conclusion that immune cell clearance is enhanced with CSPG digestion. This needs to be clarified throughout the manuscript. Fig. S1d: the difference between LV-GFP and LV-ChABC in terms of neutrophils clearance from the contused spinal cord appears small.

2. Fig. 2d: data showed opposite results at 3 dpi and 7 dpi: at 3 dpi, LV-ChABC group displayed increase score than LV-GFP (which, according to the author, would suggest increased pro-inflammatory cytokine gene expression in the ChABC treatment group), while at 7dpi, the opposite is true. How to explain this difference? Also at 14 dpi (which also represents a time point of injury resolution), there was no difference between the two groups.

3. Results are not insistent in expression changes of cytokines or inflammatory genes in response to CSPG digestion. For instance, in Fig. 3a, the authors only highlighted the genes that showed changes after CSPG digestion, but many genes did not show the anticipated changes. In addition, no significant changes were found for anti-inflammatory genes in microglia or macrophages from LV-ChABC group. Recent single cell RNA-seq studies have showed considerable heterogeneity among microglia and macrophages after SCI (Wahane et. al. Sci Adv, 2021). Hence, the pro vs. anti-inflammatory conceptual framework is an over-simplification.

4. Fig. 3e: the authors found that the inflammatory gene changes with ChABC are predominantly in the CD43 low population. What might be the mechanism? Do they express higher level of TLR4 than CD43 hi population?

5. What is the mechanism of TLR4 signaling in controlling pro-inflammatory gene expression in microglia or macrophages?

Response to reviewer comments:

Reviewer #1 (Remarks to the Author):

This study by Quijorna et al. shows that CSPGs, in addition to their well-characterized role in preventing axonal regeneration, are also involved in the recruitment of immune cells and the promotion of a proinflammatory phenotype at sites of SCI. Notably, the authors found that digestion of CSPG glycosaminoglycan side chains by means of lentivirus-mediated expression of the bacterial enzyme chondroitinase ABC (ChABC) enhanced immune cell clearance and reduced pro-inflammatory gene and protein expression profiles in the injured rat spinal cord during the subacute phase of injury. Importantly, they demonstrated that the proinflammatory effects of CSPGs are mediated through TLR4. Overall, this paper is well-written and the data are convincing using the most rigorous and unbiased methods possible. The study is relevant and important as it may contribute to explain why tissue damage is reduced and functional recovery improved in SCI rats following ChABC treatment (Bartus et al., 2014).

I only have a few minor issues with this paper, which are enumerated below and should be taken into consideration in a revised manuscript.

We thank the reviewer for these positive comments and for recognising the importance of the study. We appreciate the time taken to carefully review the manuscript and for raising several important points, which we address in full below. The additional data we provide to address these issues have considerably strengthened our manuscript and we hope you now find our paper acceptable for publication.

1) The overall message is a bit confusing when it comes to which macrophage population is affected by CSPGs in vivo (Fig. 3) versus in vitro (Fig. 5). Please also see Comment #3, below.

We appreciate this comment and address it below, together with comment #3.

2) According to Fig. 2d and Extended Figs. 3-4, although this is true that the expression of inflammatory-related genes returned to naive levels in lentivirus-ChABC treated rats compared to lentivirus-GFP treated animals at day 7 post-SCI, it is also true that signs of inflammation were higher with ChABC gene therapy at day 3. Please comment on the possibility that the temporal dynamics of inflammation and its resolution may just have been slightly shifted in time in response to ChABC treatment.

This is an interesting point. In the bar graph in Figure 2d there does indeed appear to be an elevation of inflammation in ChABC-treated animals at 3 days post injury (dpi), in comparison to lentivirus-GFP treated animals, although please note that statistically there was no significant difference between the two treatment groups (both groups showed significantly elevated pro-inflammatory cytokine expression compared to uninjured controls at 3 dpi, but were not significantly different from each other; only at 7dpi were significant changes observed between the two treatment groups). However, in order to fully address this important point (which was also raised by reviewer 3) we have performed a detailed side by side comparison of day 3 v day 7 data to directly compare the two time points and unequivocally determine if changes in cell number and inflammatory gene expression were only apparent at 7 dpi, or whether they were just shifted in time in response to ChABC treatment (see new Extended Data Figure 5). With this new analysis and additional data, we show: (i) First, a comparison of data showing the number of cells (by flow cytometry) at 3 dpi v 7 dpi. This data shows a significant reduction in response to ChABC at day 7, but not day 3, for microglia (Extended Data Figure 5a), macrophages (Extended Data Figure 5c) and neutrophils (Extended Data Figure 5e).

(ii) Second, we now provide completely new macrophage and microglia phenotype data at 3 dpi (flow cytometry of SCI tissue treated with either LV-GFP or LV-ChABC at 3 dpi, to evaluate the expression of classic M1-like and M2-like markers), to allow a side-by-side comparison of immune cell phenotype data at 3 dpi v 7 dpi (in the original manuscript we only had phenotype data at 7 and 14 dpi). We show no significant differences in immune cell phenotype markers in microglial cells

(Extended Data Figure 5b) or macrophages (Extended Data Figure 5d) at 3 dpi, but a significant decrease in pro-inflammatory markers MHCII in microglia (Extended Data Figure 5b) and iNOS, CD68 and MHCII in macrophages at 7 dpi (Extended Data Figure 5d) after lentiviral-ChABC treatment. (iii) Third, a side by side comparison of cytokine expression data (assessed in tissue at 3 dpi v 7dpi by Luminex multiplex immunoassay) shows that the only change in lentivirus-ChABC treated rats compared to GFP treatment at 3 dpi was an upregulation of leptin, which was not apparent at 7 dpi. In contrast, analysis of the prototypical pro-inflammatory cytokines CCL3, CCL5, IL1a, IL-1b and IL-18 shows no changes in lentivirus-ChABC treated rats compared to GFP treatment at 3 dpi, but a significant decrease at 7dpi (extended data Figure 5f).

(iv) Fourth, we also provide a side by side comparison of gene expression data at the two time points. Inflammatory gene expression was analysed (by qPCR), which clearly shows no changes in pro-inflammatory gene expression after lentiviral-ChABC treatment at 3 dpi compared to GFP treatment (Extended Data Figure 5g). In contrast, there was a significant reduction in numerous prototypical pro-inflammatory genes after lentiviral-ChABC treatment at 7 dpi (iNOS, CD68, MHCII, IL1b, TNFa, IL18, CCL2CCL5 and CXCL10; Extended Data Figure 5g).

We believe that with this new data and additional analysis we unequivocally demonstrate that the main effects of CSPG digestion on altering inflammatory protein and gene expression occur at 7 dpi, and are not apparent at 3 dpi, adding support to our hypothesis that the negative immune modulatory effects of CSPGs are predominantly mediated at later stages (during inflammation resolution).

Combined comments 1 and 3:

1) The overall message is a bit confusing when it comes to which macrophage population is affected by CSPGs in vivo (Fig. 3) versus in vitro (Fig. 5). Please also see Comment #3, below.

3) According to the authors' hypothesis, CSPGs that are produced during the subacute phase of SCI act on TLR4-expressing macrophages and microglia to maintain (or boost) their proinflammatory status, thus preventing inflammation resolution and maintaining a state of chronic inflammation at sites of SCI. However, the data presented in Fig. 5 suggest that CSPGs mainly act on M2 macrophages with very little effects on M1 macrophages, which seems counterintuitive. Could this have to do with the way that macrophages were polarized towards an M1 state in vitro? Indeed, M0 macrophages were turned into M1 macrophages upon treatment with LPS, a potent TLR4 agonist that leads to the activation of the LPS/CD14/TLR4 complex and its subsequent endocytosis by endosomes (this has been shown by many, including: Tan et al., *Immunity*, 2015; Rajaiah et al., *PNAS*, 2015). Could the lack of effect of CSPGs on M1 macrophages be due to their lower expression of membrane-bound TLR4 resulting from LPS-mediated TLR4 endocytosis? At the very least, TLR4 expression should be assessed in the various subsets of macrophages under investigation in culture. Thank you for pointing this out and we can understand the confusion regarding our message as to which macrophage population is most affected by CSPGs in vivo versus in vitro. We believe that the M2-like macrophage population are more affected by CSPGs than the M1-like population, and that this is true both in vitro (in BMDM cultures) and in vivo (in SCI tissue). This may seem counter-intuitive, and it may have previously been perceived that the role of CSPGs in the sub-acute phase is to maintain the proinflammatory status of macrophages and that this action would likely be on M1 macrophages. However, we show here for the first time that the main pro-inflammatory effects of CSPGs are due to the continued presence of CSPGs which block the pro-repair properties of M2-like macrophages by converting them to M1-like pro-inflammatory macrophages, and that this is mediated by the TLR4 signalling pathway. Thus, the main novelty of the paper is to show that despite CSPGs being present at earlier post-injury stages it is later, *in the resolution phase*, where they have their main detrimental effects on inflammation – activating a pro-inflammatory phenotype in pro-repair macrophages, prolonging the inflammatory response, and delaying inflammation resolution.

We now provide several pieces of new data which strengthen our evidence that M2 macrophages are more affected by CSPGs than M1 macrophages both in vitro (BMDM cultures) and in vivo (SCI injured tissue). Furthermore, we show that the differential effects are indeed likely due to

differential expression of TLR4. This is also demonstrated both *in vitro* (M1-like v M2-like BMDM cultures, rat and mouse) and *in vivo* (CD43+ and CD43- macrophage populations sorted from SCI injured tissue). The additional data is presented in two new figures (Extended Data Figures 11 and 12), which show:

(i) Firstly, we confirm that pro-inflammatory gene expression after addition of CSPGs is dramatic in M2-like BMDMs (Extended Data Figure 11b,d) in comparison to M1-like BMDMs (Extended Data Figure 11a,c); see +CSPG data (green bars) in comparison to untreated (brown dashed line) for all the data presented in Extended Data Figure 11; the rest of the data in these panels relate to additional data addressing the signalling pathways comment, point 6, below).

(ii) Secondly, in both rat and mouse studies, we demonstrate that M2-like macrophages express more TLR4 than M1-like macrophages. In polarised BMDMs from rats, we show by qPCR that RNA expression of TLR4 is 12x higher in the M2-like population than the M1-like population (Extended Data Figure 12a). We also show higher TLR4 protein expression in rat polarised M2-like BMDMs by FACS/flow cytometry (Extended Data Figure 12b,c).

(iii) We corroborate the rat data with TLR4 expression data in mice and again show higher TLR4 gene expression in the M2-like population of polarised BMDMs from mice (8 x higher than TLR4 expression in M1-like BMDMs, measured by qPCR; Extended Data Figure 12d). Protein expression in mouse BMDMs was assessed by immunocytochemistry (Extended Data Fig. 12e,f), which showed higher constitutive expression of TLR4 in M2- than M1-like BMDMs. Additionally, as the reviewer predicted, once activated by LPS M1-like macrophages internalise TLR4, evident by increased expression of TLR4 in M1-like BMDMs after permeabilization. However, despite TLR4 detection being higher in M1-like BMDMs after cell permeabilization, this did not reach the levels of TLR4 expression in M2-like BMDMs (Extended Data Fig. 12e,f).

(iv) Finally, in line with M1/M2-like BMDM data, we wanted to evaluate whether the *in vivo* stronger immunomodulatory role of CSPG in CD43-ive monocyte/macrophages compared to the CD43+ive population at 7dpi, was related with differential TLR4 expression. In sorted CD43+ive/CD43-ive macrophages derived from SCI tissue at 7dpi, we show that the CD43-ive population express significantly higher TLR4 than CD43+ive macrophages, assessed by both flow cytometry (Extended Data Fig. 12g-i) and gene expression (Extended Data Fig. 12j), which provides an explanation for why immunomodulatory effects of CSPGs are more prominent in the CD43-ive macrophage population. Together these new *in vitro* and *in vivo* data confirm our hypothesis that the main immunomodulatory effects of CSPGs are on the M2-like macrophage population. We also have altered or made additions to the text throughout to make this more clear e.g. P12, 13-14,22.

4) Please indicate whether the results suggesting neutrophil clearance following ChABC treatment through a comparison of resolution rates are significant or not (with supporting statistics).

Thank you for this suggestion. We have added statistics to the neutrophil data presented in Extended Data Figure 1d (statistics added to the legend and significance shown at 7dpi in the line graph). We also include an additional graph to show the day 7 neutrophil data alongside the day 3 neutrophil data, to more clearly show the significance only at 7dpi (new panel Extended Data Figure 5e). Furthermore, in Extended Data Fig 1d we include neutrophil data for several resolution parameters¹. This showed that Resolution Index (Ri) in neutrophils (which indicates the time needed to drop the counts of this leukocyte subset to 50% from the peak of maximal accumulation) was reduced by ~12.5h after LV-ChABC treatment (compared to LV-GFP treatment), suggesting an enhanced neutrophilic clearance. This observed quicker clearance, and a lack of significant differences in neutrophil counts at 1 dpi or 3 dpi, together suggest that is at later stages when neutrophil clearance is accelerated by CSPG digestion, thus leading to a significant reduction in neutrophil number at 7dpi. Thus, the digestion of CSPGs impacts neutrophil clearance mainly at later stages, rather than the early post-injury phase, strengthening our hypothesis of CSPGs immunomodulatory role at a late phase after SCI.

5) Can the proinflammatory effects of CSPGs on macrophages/microglia be recapitulated by biglycan alone?

This is an interesting point, given that biglycan is known to be proinflammatory and to signal through TLR4 and TLR2 in macrophages in a non-CNS model² and has been shown to be upregulated after rat contusion spinal cord injury³. It is possible, therefore, that the proinflammatory effects of CSPGs on macrophages/microglia could be predominantly due to biglycan. We have not been able to test this explicitly, since the huge amount of additional work required for this revision meant we had to prioritise our experiments. However, the CSPG mix used in our in vitro studies is a formulation that does not include biglycan (CC117, Milipore), so our working hypothesis is that biglycan is not one of the key players in CSPG-mediated modulation macrophages/microglia. This is something we would like to follow up in future studies, to characterise which specific CSPGs (or group of CSPGs) are the main mediators, and we appreciate the comment.

6) A key element that is missing are experiments that would prove that CSPG receptors other than TLR4, for example LAR and PTP σ , do not play a redundant role in this proinflammatory function of CSPGs, as suggested by others (see Dyck et al., 2018)⁴. In M1 macrophages that are only partially activated, could CSPGs elicit their proinflammatory effects via LAR or PTP σ ? Was the phosphorylation of p38 maintained in these conditions (TLR4 deficiency) as a result of CSPG stimulation?

This is an interesting and important point. To address this, we have performed additional experiments using a PTPsigma receptor inhibitor (intracellular sigma peptide, ISP), using two doses (2.5 μ M, 10 μ M) based on previously published work by other groups^{4,5}). We analysed the effects of ISP on inflammatory gene expression in M1 and M2 polarised macrophages and have performed a side by side dose comparison study with a TLR4 inhibitor (TAK-242;). The new data is presented in Extended Data Figure 11. As previously shown (Figure 6, Figure 8) CSPGs did not elicit pro-inflammatory effects in polarised M1-like macrophages (assessed by qPCR for a panel of pro-inflammatory genes; Extended Data Figure 11a,c green bars). The effects of ISP and TAK-242 in M1-polarised BMDMs were modest (TAK-242) or negligible (ISP) (Extended Data Figure 11a,c). ISP inhibition at the lowest dose elicited a small but significant increase in the M2-like marker CD206 in M1-polarised BMDMs. This is in agreement with the previous findings of Dyck et al (2018)⁴. The effects of ISP and TAK-242 on pro-inflammatory gene expression were next assessed in polarised M2-like macrophages. TLR4 inhibition with TAK-242 elicited a robust and significant reduction in pro-inflammatory gene expression in M2-like polarized macrophages at both doses. In contrast, PTP σ blockade had negligible effects in reducing CSPG-activated pro-inflammatory cytokine induction in M2-like polarized macrophages at either concentration. These findings reveal that CSPGs act predominantly via the TLR4 pathway, rather than via PTP σ , to cause M2-like polarized BMDMs to switch to a proinflammatory phenotype. These new data are presented in new Extended Data Figure 11, results P13 and discussion P22.

Regarding p38 phosphorylation in TLR4 KO BMDM. This is an interesting point and we had planned to compare the phosphorylation of p38 and its downstream molecule MAPK2 in WT and TLR4KO M2 polarized macrophages under CSPG stimulation (by Western Blot analysis). Unfortunately, it was not possible to complete these studies for this revision, due to issues with breeding and availability and ongoing restrictions for collaborative travel. However, it would be interesting to do these studies in the future.

Reviewer #2 (Remarks to the Author):

In this manuscript, Francos-Quirorna et al describes their interesting results showing that enzymatic digestion of CSPG glycosaminoglycans facilitates inflammation resolution in the spinal cord lesions. This is supported by the results from multiple non-biased and elegant analyses of immune cells in spinal cord lesions. Furthermore, they implicate TLR4 as a critical player in mediating the effects of CSPGs. Together, these results reveal a novel mechanism that regulate inflammatory response in spinal cord lesion and might have translational potential. I have only a few minor suggestions for the authors to consider during revision.

We thank the reviewer for these positive comments and for recognising the importance and novelty of the study.

While in vitro studies suggested a direct role of CSPG in regulating immune cell responses, it cannot formally rule out the possibility that CSPGs might have an indirect role. In fact, early studies showed that CSPGs could bind different soluble factors in the organization of matrix-bound cues in the brain (for example, Emerling and Lander, 17, 1089, 1996). Thus, an alternative possibility is that CSPGs sequester (pro- or anti-) inflammatory factors generated in the lesion and ChABC treatment releases these factors, contributing to the effects reported in this manuscript.

In this regard, what is the fate of glycosaminoglycans after enzymatic digestion? How long do these cleaved products remain in the lesion? Do they still have the ability of activating TLR4?

The reviewer raises several interesting and important points. To address the question regarding the potential ability of CSPG digestion products to exert immunomodulatory effects, we have performed additional experiments and provide new data. We evaluated the effect of glycosaminoglycan digestion products (Chondroitin disaccharide Δ di-0S and Δ di-4S) and compared with CSPGs on M1-like and M2-like polarized macrophages (new Extended data Fig. 10). Consistent with our previous experiments (Fig. 6) we observed that intact CSPGs cause a profound increase in pro-inflammatory gene expression in M2-like, but not M1-like, macrophages. In contrast, the disaccharides did not show any pro-inflammatory effect either in M1-like (Extended data Fig. 10a) or M2-like (Extended data Fig. 10b) polarized macrophages. Furthermore, in line with pro-repair effects of CSPG digestion, disaccharides (Chondroitin disaccharide Δ di-4S) led to increased expression of CD206 (a prototypical M2-like marker), in M2 polarized macrophages. Thus, our data showed that CSPG digestion products (Chondroitin disaccharide Δ di-0S and Δ di-4S) do not elicit pro-inflammatory activation, but rather they elicit a pro-repair phenotype, which supports our findings of the pro-resolution effects of CSPG digestion with chondroitinase. Regarding the fate and the time that these cleaved products remain in the lesion, we intend to address these important questions in future research. Please see new Extended Data Fig. 10, results P12 and additional discussion on the alternative possibility that CSPGs sequester (pro- or anti-) inflammatory factors generated in the lesion and that ChABC treatment releases these factors, contributing to the pro-resolution effects reported in this manuscript (discussion P20).

Reviewer #3 (Remarks to the Author):

Francos-Quijorna et. al. conducted detailed characterizations of the effects of CSPG digestion on the immune contexture in a rat spinal cord injury model, and attributed these changes to TLR4-mediated pro-inflammatory phenotypes of microglia/macrophages. The authors did a careful temporal comparison of the abundance of different immune cells at the injury sites using both manual flow cytometry and a t-SNE flow cytometry analysis, which showed reduced number of immune cells specifically at 7 day post-injury. They then probed inflammatory gene expression at whole tissue level and in individual cell populations. Finally, they conducted in vitro studies to show a direct effect of CSPG on upregulation of pro-inflammatory genes in M2-primed macrophages, an effect that is dependent on TLR4. Although the studies were comprehensive, the data were largely descriptive with no significant conceptual advancement for the field, and there are a number of shortcomings. We thank the reviewer for taking the time to carefully go through our manuscript. The reviewer raises several important points, which we address in full below. We respectfully disagree with the reviewer, however, that our data is largely descriptive with no significant conceptual advancement for the field. The work in our paper (now further strengthened with additional data) represents a significant conceptual advance, in several regards:

- We show for the first time that CSPGs play a critical role in preventing inflammation resolution. Impaired resolution is a critical factor in preventing tissue repair and recovery after SCI and until now little has been known about the factors involved.

- We present novel evidence that the mechanism for CSPG-mediated impaired resolution is by directly controlling immune cell phenotypic switching, and that this occurs during a key immune resolution time point.
- Phenotypic conversion of pro-repair macrophages and microglia to a pro-inflammatory phenotype by CSPGs and reversal by CSPG degradation has never previously been demonstrated.
- We also show for the first time that CSPGs can affect the adaptive immune response as well as the innate immune response, again revealing completely novel mechanistic insight into their critical role in prolonging multiple immune cell phenotypes that contribute to inflammatory pathology.
- Finally, we have discovered that Toll-like receptor 4 (TLR4) is a critical mediator of immune cell phenotypic switching, revealing a previously unknown and totally novel mechanism of action for CSPG-immune interactions.

However, we fully take on board the concerns of the reviewer and we apologise if our findings were not presented in the optimal way to appreciate their significance. We have attempted to address the concerns of the reviewer in a far as possible (without performing several years more of in vivo work and changing the focus of the study); please see responses to individual points below. We hope that the new data provided, using multiple rigorous and non-biased methods, will convince the reviewer of the merits of the study.

Major critiques:

1. Even though authors presented some new data linking CSPG and pro-inflammatory immune cell phenotype, overall the study falls short of conceptual advancement, as a number of recent studies, including PI's own work, have already shown that CSPG modulates immune response and neuroinflammation in many CNS disorders, including spinal cord injury. Along the same line, the current study has been conducted within the conceptual framework of pro- vs. anti-inflammatory immune responses, while in reality, there is considerable heterogeneity one single cell level in macrophages or microglia after CNS injury, with many cells express both anti- and pro-inflammatory genes. Hence, the role of CSPG in promoting pro-inflammatory immune responses might be an oversimplification. In fact, many inflammatory genes examined here did not consistently show changes that fit with the simplified model (see comment below).

- Regarding the perceived lack of conceptual advancement, please see our response above outlining the critical findings which demonstrate the important role of CSPGs in mediating prolonged inflammation resolution failure after SCI. These findings are now further strengthened with additional data, as outlined in our point by point responses to all three reviewers. The novel discoveries presented in our paper offer a new conceptual framework for understanding the complex non-resolving chronic inflammatory pathology of SCI.

- We agree with the reviewer that the pro v anti-inflammatory conceptual framework is somewhat of a simplification, particularly in the context of single-cell and high-plex analyses which aim to characterize novel distinct phenotypes⁶. It nevertheless represents a useful framework for summarising global changes in prototypical pro-inflammatory and anti-inflammatory associated mediators within cell populations. Accordingly this approach is widely used and currently accepted in our field^{7,8} and others across neuroscience, tissue regeneration, cancer and immunity, in recent high-impact publications⁹⁻¹³. However, we have amended our terminology throughout to make it more clear that we are using a broadly accepted classification, but we are aware that these distinctions are known to exist on a spectrum, rather than being absolute: Instead of "M1" and "M2", we use "M1-like" and "M2-like"; instead of "pro/anti-inflammatory genes/markers", we use "pro/anti-inflammatory signature genes/markers". Also, we have added to the discussion the valid point that there will be heterogeneity within individual sub-populations, but that here we are applying the pro v anti-inflammatory classification as a useful framework for looking at global changes within cell populations (discussion P17).

- The comment regarding consistent changes in the inflammatory genes examined and the "simplified model" is addressed in point 2, below.

2. It appears that authors only highlighted the results that “fit with their model”, while ignoring results that do not fit or even appear opposite. For instance, data in Fig. 2d showed opposite results at 3 dpi and 7 dpi: at 7 dpi, there was a decrease in pro-inflammatory cytokine gene expression in LV-ChABC group than in LV-GFP group, while at 3 dpi, the opposite is true. The same is also true for many pro- or anti-inflammatory genes.

We find the reviewer’s comments that refer to a “simplified model” and that we highlight only the results that “fit with our model” unmerited and unfounded. We have used multiple rigorous and unbiased methods in our study (as pointed out, and for which we were commended on, by both reviewers 1 and 2). These methods have revealed consistent changes in the inflammatory gene signatures examined, and determined that the key time point for CSPG-mediated immune modulation is 7 dpi, a critical time point for inflammation resolution after SCI.

Regarding the data in Fig. 2d and perceived differences at 3 dpi, this is a valid point (and also raised by reviewer 1). As stated above, please note that for the data referred to in the bar graph in Figure 2d, statistically there was no significant difference between the two treatment groups at 3dpi (both groups showed significantly elevated pro-inflammatory cytokine expression compared to uninjured controls at 3 dpi, but were not significantly different from each other; only at 7dpi were significant changes observed between the two treatment groups). However, we have provided additional data and analysis to address the important point of whether 7 dpi is a crucial time point for CSPG-mediated changes in pro- or anti-inflammatory signature genes/markers, or whether significant changes are observed at other post-injury time points. Please see our detailed response to reviewer 1 comment 2, above, and our new data where we have performed a detailed side by side comparison of day 3 v day 7 data to directly compare the two time points and unequivocally determine if changes in cell number and/or pro-inflammatory signature gene/marker expression were only apparent at 7 dpi, or whether significant changes (either up or down) were also apparent at 3 dpi (see new Extended Data Figure 5). With this new data and additional analysis, we unequivocally demonstrate that the main effects of CSPG digestion on altering inflammatory protein and gene expression occur at 7 dpi, and are not apparent at 3 dpi, adding support to our hypothesis that the negative immune modulatory effects of CSPGs are predominantly mediated at later stages (during inflammation resolution). See also response to additional comment 2 below, addressing effects at 14 dpi.

Additionally, we now include in Figure 2j (in vivo data) all the genes that were significantly downregulated by CSPG digestion at 7 dpi and not only the most “important” ones. We also now include in Figure 8 and Extended Data Figures 10 to 13 (in vitro data) a higher number of genes, to show effects of CSPG in BMDM polarization across all genes examined, and not only those which reached significance.

3. Another major shortcoming of the study is the lack of spatial information of the immune contexture at the injury site with or without CSPG digestion. The current study relies heavily on flow cytometry, which only revealed the abundance of different immune cells without information regarding their localization, spatial pattern, interaction with astrocytes or axons, and local matrix composition.

This is an important point, and we thank the reviewer for raising this. We provide new histological data, firstly to confirm our molecular biology findings in-situ and secondly to provide spatial context to where CSPG-immune interactions occur. We generated new in vivo tissue (adult female Lister Hooded rats, midline 150 kdyne thoracic spinal contusion injury, plus treatment with either LV-ChABC or control LV-GFP, animals perfused at 7 dpi). We performed immunohistochemistry in spinal cord tissue sections spanning the lesion site to assess the expression and distribution of two prototypical markers for M1-like and M2-like cells (iNOS and CD206, respectively) and we carried out co-localisation studies with astrocyte (GFAP) and axonal (NFH) markers as well as assessments of CSPG changes in spinal cord tissue sections at the lesion epicenter (see new Figure 4). Visualisation of intact CSPGs (observed with CS-56 expression) confirmed a reduction of CSPGs after treatment with LV-ChABC, as expected (new Figure 4a,b). Interestingly, the pattern of CSPG expression corresponded with dramatic changes in the spatial distribution of CD206+ immune cells at the injury

epicentre: in transverse spinal sections from control (LV-GFP treated) animals, we observed a ring-like pattern of CD206+ immune cells along the inner astroglial border, with CD206+ cells absent from the lesion core. In contrast, the entire lesion core was filled with CD206+ cells in the spinal cords of LV-ChABC treated animals. The densely packed core of CD206+ immune cells was mirrored (almost exactly) by an absence of CSPGs: compare panels in Fig. a,c (dense CSPG in core, CD206+ cells excluded from the core) with panels in Fig. b,d (absence of CSPG in core, abundance of CD206+ cells in the core). Furthermore, the increased CD206 expression in LV-ChABC treated animals was also mirrored by a reduction in the M1-like marker iNOS both within the lesion core and border (where iNOS expression appeared to be associated with areas of injury-induced loss of NFH). These findings support our hypothesis that CSPGs play a major role in influencing immune cell phenotype. Notably, when CSPGs are degraded with ChABC, the injury core is no longer filled with toxic immune mediators such as iNOS but is abundant in pro-repair macrophages. **We therefore hypothesise that CSPGs prevent immune cells at the very injury core from converting to a pro-repair phenotype, and this leads to tissue necrosis at the injury epicentre and subsequent tissue cavitation.** Moreover, the observation of the ring-like pattern of CD206+ immune cells around the inner astroglial border raises the possibility that these cells could be linked to the “microglial scar” recently proposed by Lacroix¹⁴, which are localised at the interface between the astroglial scar border and the inner lesion core, and this seems a promising new avenue to explore in future studies. Our new data is presented in new Figure 4 and results Px-x “Differences in immune cell spatial distribution after CSPG digestion”, and we include an additional discussion section on how the observed spatial localisation and pattern provides insight into how CSPG-mediated immunomodulation may directly contribute to chronic tissue pathology and cavitation and how it may impact the composition of the spinal injury scar (including interactions with astroglial and microglial scar components). We believe this data adds a new dimension to our study, and opens up interesting avenues for future research on ECM-immune-scar interactions, and we thank the reviewer for requesting this additional data. neurons (see new Figure 4, results P8-9, discussion P18-19).

4. There is also a missed opportunity to compare axon regeneration and functional recovery in TLR4-/- mutant mice vs. mice treated with LV-ChABC. If TLR4 functions mainly as a receptor for CSPG in promoting pro-inflammatory phenotypes of microglia/macrophages, TLR4 KO would phenocopy LV-ChABC. More interesting perhaps is to conduct SCI study combining TLR4 KO and LV-ChABC, which would will provide crucial in vivo genetic evidence linking CSPG and TLR4. It will be interesting to determine if combination would result in no further gain in reducing inflammatory milieu, although CSPG may function through other receptors or on other cells such as astrocytes.

As reviewer #3 points out, one of our main findings is that TLR4 functions as a main receptor for CSPG in promoting pro-inflammatory phenotypes of microglia/macrophages. However, we do not think that in vivo TLR4 KO would phenocopy the LV-ChABC effect. TLR4 is necessary to develop a proper and necessary inflammatory response (and indeed is a mechanism to recover the homeostasis of the tissue). Thus KO of TLR4 will affect the inflammatory reaction in multiple different ways which could either be beneficial or detrimental (since TLR4 is expressed in many cell types and has affinity with many ligands which have both positive and detrimental effects in the inflammatory response). Indeed, previous work has shown conflicting evidence regarding in vivo TLR4-/- after SCI, with reports by different groups demonstrating improved¹⁵ or worsened¹⁶⁻¹⁸ functional outcome in TLR4 KO mutant mice. These differing effects are likely dependent on the time when KO is induced, with detrimental effects if deleted at the beginning of initial necessary inflammation, but representing a good therapeutic target if deleted in later stages, to avoid chronification of the inflammatory response, as seen with CSPG signalling. Nevertheless, we agree with the reviewer that it could be interesting to test whether the combination of LV-ChABC treatment with macrophage-dependent TLR4 deletion would synergize and result in improved functional recovery. However, this experiment would require a conditional deletion of TLR4 restricted to the myeloid line only, inducible after the very early inflammatory stages, in order to not interfere with required early inflammatory activation and other, non-inflammatory cell TLR4-dependent signalling. To do so, we would need to generate a new line, for instance by crossing the

conditional knockout TLR4^{fl} line (<https://www.jax.org/strain/024872>) and the tamoxifen cre-dependent line for myeloid cell specific deletion (<https://www.jax.org/strain/031674>). To generate these mice and then evaluate axonal regeneration and functional outcomes in these conditions would require at least one more year of work (and produce an additional manuscript worth of data). Importantly, as we mentioned before, the main aim of the work presented here was to evaluate the immunomodulatory role of CSPG after SCI and the signalling pathways that mediate this, and not axonal regeneration. Thus, while this could be a useful avenue to pursue in the future, we feel it is out of the scope of the present study and would significantly change the focus of the paper.

However, we have attempted to address the point regarding axon regeneration in TLR4^{-/-} mutant mice vs mice treated with LV-ChABC in a reduced neuronal culture model. We carried out a series of in vitro studies to look at neuronal responses to CSPGs +/- ChABC +/- TLR4 KO. We carried out neurite outgrowth assays using cultured neurons from TLR4 mutant mice and WT controls (cultured cortical neurons from E14.5 WT or TLR4^{-/-} mice; 3 days after plating CSPG or vehicle were added +/- ChABC for 24 hours; neurite outgrowth assessed by β -III-tubulin immunocytochemistry). As expected, and in line with numerous reports in the literature¹⁹⁻²², addition of CSPGs inhibited neurite outgrowth and digestion of CSPGs with ChABC promoted neurite outgrowth. We show that the effects of CSPGs on neuronal growth are not dependent on TLR4 signalling, since similar effects of CSPG activation or digestion were observed in both WT and TLR4^{-/-} neurons (see new Extended Data Figure 15, results P15-16, discussion P22). Thus, the well known effects of CSPGs on blocking regenerative growth are not mediated by an interaction of CSPGs with TLR4. The main role of CSPG-TLR4 interactions after SCI therefore appears to be modulation of the immune cell response to injury. Finally, whether CSPGs may function through other receptors is addressed above (see response to reviewer 1, point 6).

Additional comments:

1. Fig. 1: After SCI, macrophages continue to arrive at the injury site, while early-arrived macrophages are being cleared. Hence the reduced number of macrophages at 7 dpi may also reflect a reduction of new recruitment of monocytes/macrophages, and not necessarily clearance. Without lineage tracing experiments, the data do not support the conclusion that immune cell clearance is enhanced with CSPG digestion. This needs to be clarified throughout the manuscript. Fig. S1d: the difference between LV-GFP and LV-ChABC in terms of neutrophils clearance from the contused spinal cord appears small.

As suggested, we have amended the text throughout to state that the reduced number of macrophages at 7 dpi may reflect either enhanced immune cell clearance (which is our hypothesis) or a reduction of new recruitment of monocytes/macrophages. Regarding the significance of the neutrophil data (Extended data 1d) please see response to Reviewer 1 point 4 and new panel in Extended Data Figure 5e.

2. Fig. 2d: data showed opposite results at 3 dpi and 7 dpi: at 3 dpi, LV-ChABC group displayed increase score than LV-GFP (which, according to the author, would suggest increased pro-inflammatory cytokine gene expression in the ChABC treatment group), while at 7dpi, the opposite is true. How to explain this difference? Also at 14 dpi (which also represents a time point of injury resolution), there was no difference between the two groups.

Regarding 3 dpi vs 7 dpi data, see response to Major Point 2 above which addresses this concern (and see new Extended Data Figure 5).

Regarding 14 dpi, there were some differences between the two groups still apparent, although to a lesser extent than at 7 dpi. Namely:

- Reduced MHCII in microglia and macrophages was maintained at 14 dpi. See Extended Data Fig. 6b,c (previously Extended Data Fig. 5a,b).

- Reduced CD68 expression was observed in sorted microglial cells at 14 dpi. See new Extended Data Fig. 6d.

- A reduction of some inflammatory cytokines was still observed in lesioned tissue at 14 dpi (assessed by qPCR), although to a lesser extent than 7dpi. See new Extended Data Fig. 6a, which shows a side by side comparison of 7 dpi and 14 dpi.

Thus, although the predominant effects of CSPG-mediated immune modulation occur at 7 dpi (a key resolution time point after SCI), there are still some effects apparent at 14 dpi (a time point where resolution is still occurring but is diminishing).

3. Results are not insistent in expression changes of cytokines or inflammatory genes in response to CSPG digestion. For instance, in Fig. 3a, the authors only highlighted the genes that showed changes after CSPG digestion, but many genes did not show the anticipated changes. In addition, no significant changes were found for anti-inflammatory genes in microglia or macrophages from LV-ChABC group. Recent single cell RNA-seq studies have showed considerable heterogeneity among microglia and macrophages after SCI (Wahane et. al. Sci Adv, 2021). Hence, the pro vs. anti-inflammatory conceptual framework is an over-simplification.

Apologies if there was any confusion. The data in Figure 3a-h is not gene expression data, but evaluation by flow cytometry markers linked with M1 and M2-like polarization (markers that are accepted worldwide as the most useful markers to evaluate macrophage/microglial polarization). Thus, this is not the same experiment as that shown in Figure 2, is not showing RNA data, or highlighting any specific data selected from gene expression studies. The reviewer is correct in that although there was a trend for increased expression of anti-inflammatory M2-like markers with LV-ChABC, this was not statistically significant and we have modified the text to make this more clear (px). On the other hand, the significant reduction of M1-like markers clearly demonstrate the immunomodulatory role of CSPG in macrophage/microglial cell polarization. Finally, this modulation in macrophage and microglial cell polarization by CSPGs is further corroborated by gene expression data in Figure 3k in sorted microglia and macrophages. See also our responses to major comments 1 and 2 above, regarding consistency in our findings relating to expression changes of cytokines or inflammatory genes in response to CSPG activation and digestion.

Finally, in addition to demonstrating differential effects of CSPG stimulation on macrophage gene expression, we also provide new data to address the question of how CSPG stimulation affects macrophage function/ behaviour. Phagocytosis is an important mechanism to recover tissue homeostasis by which macrophages remove injury-induced cellular and environmental debris. In an assay which reflects the ability of macrophages to perform phagocytosis, we observed that M2-like macrophages, but not M1-like macrophages, phagocytose significantly less when stimulated with CSPGs (see new Extended Data Fig.8). Thus, at both a gene expression level, and a functional level, CSPG stimulation makes M2-like macrophages more similar to M1-like macrophages.

4. Fig. 3e: the authors found that the inflammatory gene changes with ChABC are predominantly in the CD43 low population. What might be the mechanism? Do they express higher level of TLR4 than CD43 hi population?

This is an important point. Please see response to reviewer 1, comment 3 above and new TLR4 expression data (new Extended Data Figure 12).

5. What is the mechanism of TLR4 signaling in controlling pro-inflammatory gene expression in microglia or macrophages?

This is an important point which we have addressed with previous and completely new data in Extended Data Figure 12. To assess TLR4 signalling by CSPG we evaluated by Western Blot the phosphorylation of two main pathways related with TLR4 activation, MAPK and NfκB. Our results showed that p38 was significantly more phosphorylated in M2-like polarized rat BMDM after 4h and 16h CSPG treatment, compared with untreated controls (Extended Data Figure 12a-d; previously Extended Data Figure 7). This result was corroborated in mouse BMDMs by fluorescence intensity of p-p38 (new Extended Data Figure 12e-f). Finally, a p38 inhibitor (SB202190) significantly reduced several pro-inflammatory genes that were upregulated after CSPG stimulation (IL1b, IL6, CXCL10) in M2-like polarized rat BMDMs (new Extended Data Figure 12g-h). These results suggest that the

TLR4-dependent immunomodulatory role exhibited by CSPG on M2-like macrophages and microglial cells is mediated, at least in part, by activation of the MAPK pathway. More specifically, we suggest that activating p38 phosphorylation and its translocation to the nucleus promotes an upregulation of pro-inflammatory cytokines such as IL1b, IL6 and CXCL10.

References:

- 1 Prüss, H. *et al.* Non-Resolving Aspects of Acute Inflammation after Spinal Cord Injury (SCI): Indices and Resolution Plateau. *Brain Pathology* **21**, 652-660, doi:<https://doi.org/10.1111/j.1750-3639.2011.00488.x> (2011).
- 2 Schaefer, L. *et al.* The matrix component biglycan is proinflammatory and signals through Toll-like receptors 4 and 2 in macrophages. *The Journal of Clinical Investigation* **115**, 2223-2233, doi:10.1172/JCI23755 (2005).
- 3 Didangelos, A. *et al.* High-throughput proteomics reveal alarmins as amplifiers of tissue pathology and inflammation after spinal cord injury. *Sci Rep* **6**, 21607 doi: 10.1038/srep21607 (2016).
- 4 Dyck, S. *et al.* Perturbing chondroitin sulfate proteoglycan signaling through LAR and PTP σ receptors promotes a beneficial inflammatory response following spinal cord injury. *J Neuroinflammation* **15**, 90-90, doi:10.1186/s12974-018-1128-2 (2018).
- 5 Tran, A. P., Sundar, S., Yu, M., Lang, B. T. & Silver, J. Modulation of Receptor Protein Tyrosine Phosphatase Sigma Increases Chondroitin Sulfate Proteoglycan Degradation through Cathepsin B Secretion to Enhance Axon Outgrowth. *J Neurosci* **38**, 5399-5414, doi:10.1523/jneurosci.3214-17.2018 (2018).
- 6 Dumas, A. A., Borst, K. & Prinz, M. Current tools to interrogate microglial biology. *Neuron* **109**, 2805-2819, doi:10.1016/j.neuron.2021.07.004 (2021).
- 7 David, S. & Kroner, A. Repertoire of microglial and macrophage responses after spinal cord injury. *Nature Reviews Neuroscience* **12**, 388-399, doi:10.1038/nrn3053 (2011).
- 8 Kigerl, K. A. *et al.* Identification of Two Distinct Macrophage Subsets with Divergent Effects Causing either Neurotoxicity or Regeneration in the Injured Mouse Spinal Cord. *The Journal of Neuroscience* **29**, 13435-13444, doi:10.1523/jneurosci.3257-09.2009 (2009).
- 9 van der Vlist, M. *et al.* Macrophages transfer mitochondria to sensory neurons to resolve inflammatory pain. *Neuron*, doi:10.1016/j.neuron.2021.11.020 (2021).
- 10 Hu, G. *et al.* High-throughput phenotypic screen and transcriptional analysis identify new compounds and targets for macrophage reprogramming. *Nat Commun* **12**, 773, doi:10.1038/s41467-021-21066-x (2021).
- 11 Wei, Z. *et al.* Boosting anti-PD-1 therapy with metformin-loaded macrophage-derived microparticles. *Nat Commun* **12**, 440, doi:10.1038/s41467-020-20723-x (2021).
- 12 Shang, M. *et al.* Macrophage-derived glutamine boosts satellite cells and muscle regeneration. *Nature* **587**, 626-631, doi:10.1038/s41586-020-2857-9 (2020).
- 13 He, L. *et al.* Global characterization of macrophage polarization mechanisms and identification of M2-type polarization inhibitors. *Cell Rep* **37**, 109955, doi:10.1016/j.celrep.2021.109955 (2021).
- 14 Bellver-Landete, V. *et al.* Microglia are an essential component of the neuroprotective scar that forms after spinal cord injury. *Nature Communications* **10**, 518, doi:10.1038/s41467-019-08446-0 (2019).
- 15 Xu, S. *et al.* TLR4 promotes microglial pyroptosis via lncRNA-F630028O10Rik by activating PI3K/AKT pathway after spinal cord injury. *Cell Death & Disease* **11**, 693, doi:10.1038/s41419-020-02824-z (2020).
- 16 Church, J. S., Kigerl, K. A., Lerch, J. K., Popovich, P. G. & McTigue, D. M. TLR4 Deficiency Impairs Oligodendrocyte Formation in the Injured Spinal Cord. *J Neurosci* **36**, 6352-6364, doi:10.1523/JNEUROSCI.0353-16.2016 (2016).

- 17 Kigerl, K. A. *et al.* Toll-like receptor (TLR)-2 and TLR-4 regulate inflammation, gliosis, and myelin sparing after spinal cord injury. *Journal of Neurochemistry* **102**, 37-50, doi:<https://doi.org/10.1111/j.1471-4159.2007.04524.x> (2007).
- 18 Impellizzeri, D. *et al.* Role of Toll like receptor 4 signaling pathway in the secondary damage induced by experimental spinal cord injury. *Immunobiology* **220**, 1039-1049, doi:<https://doi.org/10.1016/j.imbio.2015.05.013> (2015).
- 19 Stern, S. *et al.* RhoA drives actin compaction to restrict axon regeneration and astrocyte reactivity after CNS injury. *Neuron* **109**, 3436-3455.e3439, doi:10.1016/j.neuron.2021.08.014 (2021).
- 20 Wu, D. *et al.* Chronic neuronal activation increases dynamic microtubules to enhance functional axon regeneration after dorsal root crush injury. *Nat Commun* **11**, 6131, doi:10.1038/s41467-020-19914-3 (2020).
- 21 Lang, B. T. *et al.* Modulation of the proteoglycan receptor PTP α promotes recovery after spinal cord injury. *Nature* **518**, 404-408, doi:10.1038/nature13974 (2015).
- 22 Fidler, P. S. *et al.* Comparing astrocytic cell lines that are inhibitory or permissive for axon growth: the major axon-inhibitory proteoglycan is NG2. *J Neurosci* **19**, 8778-8788, doi:10.1523/jneurosci.19-20-08778.1999 (1999).

Reviewers' Comments:

Reviewer #1:

Remarks to the Author:

This is a resubmission of a manuscript first submitted in Spring of 2021. The authors have taken very seriously the suggestions and comments made by the three reviewers, and now present a substantially revised manuscript in which all major critiques were satisfactorily addressed. In particular, the authors have added new data in which they unequivocally demonstrate the main effects of CSPG digestion on altering inflammatory gene and protein expression (Extended Data Fig. 5), as well as provided new evidence showing that M2 macrophages are the main immune cells targeted by CSPGs due to their strong TLR4 expression (Extended Data Figs. 11-12). The authors should be commended for their efforts to resolve these important issues and address the most relevant comments.

Reviewer #2:

Remarks to the Author:

In this revised manuscript, the authors have appropriately addressed all issues i raised previously. It is suitable for publication now.

Reviewer #3:

Remarks to the Author:

The authors addressed all critiques satisfactorily in the revision.

one typo: line 287: should be cytotoxic T lymphocytes (TCD8).